# Two-Stage Predict+Optimize for Mixed Integer Linear Programs with Unknown Parameters in Constraints

**Xinyi Hu[1], Jasper C.H. Lee[2], Jimmy H.M. Lee[1]**
[1]Department of Computer Science and Engineering
The Chinese University of Hong Kong, Shatin, N.T., Hong Kong
[2]Department of Computer Sciences & Institute for Foundations of Data Science
University of Wisconsin–Madison, WI, USA
{xyhu,jlee}@cse.cuhk.edu.hk, jasper.lee@wisc.edu

## Abstract

Consider the setting of constrained optimization, with some parameters unknown at solving time and requiring prediction from relevant features. Predict+Optimize is a recent framework for end-to-end training supervised learning models for such predictions, incorporating information about the optimization problem in the training process in order to yield better predictions in terms of the quality of the predicted solution under the true parameters. Almost all prior works have focused on the special case where the unknowns appear only in the optimization objective and not the constraints. Hu et al. proposed the first adaptation of Predict+Optimize to handle unknowns appearing in constraints, but the framework has somewhat ad-hoc elements, and they provided a training algorithm only for covering and packing linear programs. In this work, we give a new *simpler* and *more powerful* framework called *Two-Stage Predict+Optimize*, which we believe should be the canonical framework for the Predict+Optimize setting. We also give a training algorithm usable for all mixed integer linear programs, vastly generalizing the applicability of the framework. Experimental results demonstrate the superior prediction performance of our training framework over all classical and state-of-the-art methods.

## 1 Introduction

Optimization problems are prevalent in modern society, and yet the problem parameters are not always available at the time of solving. For example, consider the real-world application scenario of stocking a store: as store managers, we need to place monthly orders for products to stock in the store. We want to stock products that sell fast and yield high profits, as much of them as possible, subject to the hard constraint of limited storage space. However, orders need to be placed two weeks in advance of the monthly delivery, and the customer demand next month cannot be known exactly at the time of order placement. In this paper, we consider the supervised learning setting, where the unknown parameters can be predicted from relevant features, and there are sufficient historical (features, parameters) pairs as training data for a prediction model. The goal, then, is to learn a prediction model from the training data such that, if we plug in the estimated parameters into the optimization problem and solve for an *estimated solution*, the estimated solution remains a good solution even after the true parameters are revealed.

The classic approach to the problem would be to train a simple regression model, based on standard losses such as (regularized) $\ell_2$ loss, to predict parameters from the features. It is shown, however, that having a small prediction error in the parameter space does not necessarily mean that the estimated solution performs well under the true parameters. The recent framework of Predict+Optimize, by

Elmachtoub and Grigas [7], instead proposes the more effective *regret* loss for training, which compares the solution qualities of the true optimal solution and the estimated solution under the true parameters. Subsequent works [6, 8, 10, 13, 17, 19, 27] have since appeared in the literature, applying the framework to more and wider classes of optimization problems as well as focusing on speed-vs-prediction accuracy tradeoffs.

However, all these prior works focus only on the case where the unknown parameters appear in the optimization objective, and not in the constraints. The technical challenge for the generalization is immediate: if there were unknown parameters in the constraints, the estimated solution might not even be feasible under the true parameters revealed afterwards! Thus, in order to tackle the Predict+Optimize setting with unknowns in constraints, the recent work of Hu et al. [12] presents the first such adaptation on the framework: they view the estimated solution as representing a *soft commitment*. Once the true parameters are revealed, corrective action can be taken to ensure feasibility, potentially at a penalty corresponding to the real-life cost of (partially) reneging on a soft commitment. Their framework captures application scenarios whenever such correction is possible, and requires the practitioner to specify both the correction mechanism and the penalty function. These data can be determined and derived from the specific application scenario. As an example, in the product-stocking problem, an additional unknown parameter is the storage space, because it depends on how the current products in the store sell before the new order arrives. We need to place orders two weeks ahead based on predicted storage space. The night before the order arrives, we know the precise available space, meaning that the unknown parameter is revealed. A possible correction mechanism then is to throw away excess products that the store cannot keep, while incurring the penalty that is the retail price of the products, as well as disposal fees.

While the Hu et al. [12] framework does capture many application scenarios, there are important shortcomings. In their framework, they require the practitioner to specify a correction function that amends an infeasible solution into a feasible solution. However, the derivation of a correction function can be rather ad-hoc in nature. In particular, given an infeasible estimated solution, there may be many ways to transform the solution into a feasible one, and yet their framework requires the practitioner to pick one particular way. This leads to the second downside: it is difficult to give a *general* algorithmic framework that applies to a wide variety of optimization problems. Hu et al. had to restrict their attention only to packing and covering linear programs, for which they could propose a generic correction function. In this work, we aim to *vastly generalize* the kinds of optimization problems that Predict+Optimize can tackle under uncertainty in the constraints. In addition, the approach of Hu et al. fails to handle the interesting situation in which post-hoc correction is still desirable when the estimated solution is feasible but not good under the true parameters.

Our contributions are three-fold:

• To mitigate the shortcomings of the prior work, we propose and advocate a new framework, which we call *Two-Stage Predict+Optimize*[1], that is both *conceptually simpler* and *more expressive* in terms of the class of optimization problems it can tackle. The key idea for the new framework is that the correction function is unnecessary. All that is required is a penalty function that captures the cost of modifying one solution to another. A penalty function is sufficient for defining a correction process: we formulate the correction process itself as a "Stage 2" optimization problem, taking the originally estimated solution as well as the penalty function into account.

• Under this framework, we further propose a general end-to-end training algorithm that applies not only to packing and covering linear programs, but also to all mixed integer linear programs (MILPs). We adapt the approach of Mandi and Guns [18] to give a gradient method for training neural networks to predict parameters from features.

• We apply the proposed method to three benchmarks to demonstrate the superior empirical performance over classical and state-of-the-art training methods.

## 2   Background

In this section, we give basic definitions for optimization problems and the Predict+Optimize setting [7], and describe the state-of-the-art framework [12] for Predict+Optimize with unknown parameters

---

[1] The literature sometimes uses "two-stage" to mean approaches where the prediction is agnostic to the optimization problem. Here, "two-stage" refers to the soft commitment and the correction.

in constraints. The theory is stated in terms of minimization but applies of course also to maximization, upon appropriate negation. Without loss of generality, an *optimization problem* (OP) $P$ can be defined as finding:

$$x^* = \arg\min_x obj(x) \quad \text{s.t. } C(x)$$

where $x \in \mathbb{R}^d$ is a vector of decision variables, $obj : \mathbb{R}^d \to \mathbb{R}$ is a function mapping $x$ to a real objective value that is to be minimized, and $C$ is a set of constraints that must be satisfied over $x$. We call $x^*$ an *optimal solution* and $obj(x^*)$ the *optimal value*. A *parameterized optimization problem (Para-OP)* $P(\theta)$ is an extension of an OP $P$:

$$x^*(\theta) = \arg\min_x obj(x, \theta) \quad \text{s.t. } C(x, \theta)$$

where $\theta \in \mathbb{R}^t$ is a vector of parameters. The objective $obj(x, \theta)$ and constraints $C(x, \theta)$ can both depend on $\theta$. When the parameters are known, a Para-OP is just an OP.

In the *Predict+Optimize* setting [7], the true parameters $\theta \in \mathbb{R}^t$ for a Para-OP are not known at solving time, and *estimated parameters* $\hat{\theta}$ are used instead. Suppose each parameter is estimated by $m$ features. The estimation will rely on a machine learning model trained over $n$ observations of a training data set $\{(A^1, \theta^1), \ldots, (A^n, \theta^n)\}$ where $A^i \in \mathbb{R}^{t \times m}$ is a *feature matrix* for $\theta^i$, in order to yield a *prediction function* $f : \mathbb{R}^{t \times m} \to \mathbb{R}^t$ predicting parameters $\hat{\theta} = f(A)$.

Solving the Para-OP using the estimated parameters, we obtain an *estimated solution* $x^*(\hat{\theta})$. When the unknown parameters appear in constraints, one major challenge is that the feasible region is only approximated at solving time, and hence the estimated solution may be infeasible under the true parameters. Fortunately, in certain applications, the estimated solution is not a hard commitment, but only represents a soft commitment that can be modified once the true parameters are revealed. Hu et al. [12] propose a Predict+Optimize framework for such applications. The framework is as follows: i) the unknown parameters are estimated as $\hat{\theta}$, and an estimated solution $x^*(\hat{\theta})$ is solved using the estimated parameters, ii) the true parameters $\theta$ are revealed, and if $x^*(\hat{\theta})$ is infeasible under $\theta$, it is *amended* into a *corrected solution* $x^*_{\text{corr}}(\hat{\theta}, \theta)$ while potentially incurring some *penalty*, and finally iii) the solution $x^*_{\text{corr}}(\hat{\theta}, \theta)$ is evaluated according to the sum of both the objective, under the true parameters $\theta$, and the incurred penalty from correction.

More formally, a *correction function* takes an estimated solution $x^*(\hat{\theta})$ and true parameters $\theta$ and returns a *corrected solution* $x^*_{corr}(\hat{\theta}, \theta)$ that is feasible under $\theta$. A *penalty function* $Pen(x^*(\hat{\theta}) \to x^*_{corr}(\hat{\theta}, \theta))$ takes an estimated solution $x^*(\hat{\theta})$ and the corrected solution $x^*_{corr}(\hat{\theta}, \theta)$ and returns a non-negative penalty. Both the correction function and the penalty function should be chosen according to the precise application scenario at hand. The final corrected solution $x^*_{corr}(\hat{\theta}, \theta)$ is evaluated using the *post-hoc regret*, which is defined with respect to the corrected solution $x^*_{corr}(\hat{\theta}, \theta)$ and the penalty function $Pen(x^*(\hat{\theta}) \to x^*_{corr}(\hat{\theta}, \theta))$. The post-hoc regret is the sum of two terms: (a) the difference in objective between the *true optimal solution* $x^*(\theta)$ and the corrected solution $x^*_{corr}(\hat{\theta}, \theta)$ under true parameters $\theta$, and (b) the penalty that the correction process incurs. Mathematically, the post-hoc regret function $PReg(\hat{\theta}, \theta) : \mathbb{R}^t \times \mathbb{R}^t \to \mathbb{R}_{\geq 0}$ (for minimization problems) is:

$$PReg(\hat{\theta}, \theta) = obj(x^*_{corr}(\hat{\theta}, \theta), \theta) - obj(x^*(\theta), \theta) + Pen(x^*(\hat{\theta}) \to x^*_{corr}(\hat{\theta}, \theta)) \tag{1}$$

where $obj(x^*_{corr}(\hat{\theta}, \theta), \theta)$ is the *corrected optimal value* and $obj(x^*(\theta), \theta)$ is the *true optimal value*.

Given the post-hoc regret as a loss function, the empirical risk minimization principle dictates that we choose the prediction function to be the function $f$ from the set of models $\mathcal{F}$ attaining the smallest average post-hoc regret over the training data:

$$f^* = \arg\min_{f \in \mathcal{F}} \frac{1}{n} \sum_{i=1}^{n} PReg(f(A^i), \theta^i) \tag{2}$$

## 3 Two-stage Predict+Optimize Framework

While the prior work by Hu et al. [12] is the first Predict+Optimize framework for unknowns in constraints, and is indeed applicable to a good range of applications, it has several shortcomings.

First, the framework requires mathematically formalizing both a penalty function and a correction function from the application scenario, and essentially imposes differentiability assumptions on the correction function for the framework to be usable. The ad-hoc nature of writing down a correction function limits the practical applicability of the framework. Second, as a result of needing a single (differentiable) correction function, Hu et al. [12] needed to restrict their attention to only packing and covering linear programs, in order to derive a general correction function that is applicable to all the instances. This also significantly limits the immediate applicability of their framework. Third, their framework only corrects an estimated solution when it is infeasible under the true parameters. Yet, there are applications where corrections are possible even when the estimated solution were feasible, but just not very good under the true parameters.

In this paper, we advocate using a simpler yet more powerful framework, which we call *Two-Stage Predict+Optimize*, addressing all of the above shortcomings. The simplified perspective will allow us to discuss more easily how to handle the entire class of mixed integer linear programs (MILPs) instead of being restricted to just packing and covering linear programs. Since MILPs include all optimization problems in NP (under a reasonable definition of NP for optimization problems), our framework is significantly more applicable in practice. We will describe the Two-Stage Predict+Optimize framework below, and discuss its application to MILPs in the next section.

Our framework is simple: we forgo the idea of a correction function and treat correction itself as an optimization problem, based on the penalty function, the estimated solution and the revealed true parameters. Recall the Hu et al. view of Predict+Optimize under uncertainties in constraints: the estimated solution is a form of soft commitment, which can be modified at a cost once the true parameters are revealed. The penalty function describes the cost of changing from an estimated solution to a final solution. The main observation is that, given an estimated solution and the revealed parameters, we should in fact solve a *new* optimization problem, formed by applying the true parameters to the original optimization, and adding the penalty function to the objective. The final solution from this new optimization thus takes the penalty of correction into account. This approach yields three immediate advantages. First, the practitioner no longer needs to specify a correction function, thus reducing the ad-hoc nature of the framework. Second, even feasible solutions are allowed to be modified after the true parameters are revealed if the penalty of doing so is not infinity. Third, conditioned on the same penalty function, the solution quality from our two-stage optimization approach is always at least as good as that from using any correction function. The last advantage is presented as Proposition A.1.

Now we formally define the Two-Stage Predict+Optimize framework.

**I.** In Stage 1, the unknown parameters are estimated as $\hat{\theta}$ from features. The practitioner then solves the *Stage 1* optimization, which is the Para-OP using the estimated parameters, to obtain the *Stage 1 solution* $x_1^*$. The Stage 1 solution should be interpreted as some form of soft commitment, that we get to modify in Stage 2 at extra cost/penalty. Assuming the notation of the Para-OP in Section 2, the Stage 1 OP can be formulated as:

$$x_1^* = \arg\min_x \ obj(x, \hat{\theta}) \quad \text{s.t.} \ C(x, \hat{\theta})$$

**II.** At the beginning of Stage 2, the true parameters $\theta$ are revealed. The Stage 2 optimization problem augments the original Stage 1 problem by adding a penalty term $Pen(x_1^* \to x_2^*, \theta)$ to the objective, which accounts for the penalty (modelled from the application scenario) for changing from the softly-committed Stage 1 solution $x_1^*$ to the new *Stage 2* and *final* solution $x_2^*$. The Stage 2 OP can then be formulated as:

$$x_2^* = \arg\min_x \ obj(x, \theta) + Pen(x_1^* \to x, \theta) \quad \text{s.t.} \ C(x, \theta)$$

Solving the Stage 2 problem yields the final Stage 2 "corrected" solution $x_2^*$.

**III.** The Stage 2 solution $x_2^*$ is evaluated according to the analogous post-hoc regret, as follows:

$$PReg(\hat{\theta}, \theta) = obj(x_2^*, \theta) + Pen(x_1^* \to x_2^*, \theta) - obj(x^*(\theta), \theta)$$

where again, $x^*(\theta)$ is an optimal solution of the Para-OP under the true parameters $\theta$. Note that the post-hoc regret depends on *all* of a) the predicted parameters, b) the induced Stage 1 solution, c) the true parameters and d) the final Stage 2 solution.

To see this new framework applies in practice, the following example expands on the product-stocking problem in the introduction.

**Example 1.** *Consider the product-stocking problem again, where regular orders have to be placed two weeks ahead of monthly deliveries. Since the available space at the time of delivery is unknown when we place the regular orders, depending on the sales over the next two weeks, we need to make a prediction on the available space to make a corresponding order. We learn the predictor using historical sales records from features such as time-of-year and price. Then, we use the predicted available space to optimize for the regular order we place. This is the Stage 1 solution.*

*The night before the order arrives, the unknown constraint parameter, i.e. the precise available space, is revealed. We can then check if we have over-ordered or under-ordered. In the case of over-ordering, we would have to call and ask the wholesale company to drop some items from the order. The company would perhaps allow taking the items off the final bill, but naturally they have a surcharge for last-minute changes. Similarly, if we under-ordered, we might request the wholesale company to send us more products, again naturally with a surcharge for last-minute ordering. The updated order is the Stage 2 decision. The incurred wholesaler surcharges induce the penalty function.*

A reader who is familiar with the literature on two-stage optimization problems may note that the above framework is phrased slightly differently from some other two-stage problem formulations. In particular, some two-stage frameworks phrase Stage 1 solutions as *hard* commitments, and include *recourse variables* in both stages of optimization to model what changes are made in Stage 2. We show in Appendix A.1 how our framework can capture this other perspective, and in general discuss how problem modelling can be done in our new framework.

The reader may also wonder: what about application scenarios where the (Stage 1) estimated solution is a hard commitment, and there is absolutely no correction/recourse available? In Appendix A.2, we discuss how our framework is *still* useful and applicable for learning in these situations.

We also give a more detailed comparison, in Appendix A.3, between our new Two-Stage Predict+Optimize framework and the prior Hu et al. framework. Technically, if we *ignored* differentiability issues, the two frameworks are mathematically equivalent in expressiveness. However, we stress that our new framework is both *conceptually simpler* and *easier to apply* to a *far wider* class of optimization problems. We show concretely in the next section how to end-to-end train a neural network for this framework for all MILPs, vastly generalizing the method of Hu et al. which is restricted to packing and covering (non-integer) linear programs. In addition, Appendix A.3 also states and proves Proposition A.1, that if we fix an optimization problem, a prediction model and a penalty function, then the solution quality from our two-stage approach is always at least as good as using the correction function approach.

## 4 Two-Stage Predict+Optimize on MILPs

In this section, we describe how to give an end-to-end training method for neural networks to predict unknown parameters from features, under the Two-Stage Predict+Optimize framework. The following algorithmic method is applicable whenever *both* stages of optimization are expressible as MILPs. Due to the page limit, the discussion in this section is high-level and brief, with all the calculation details deferred to Appendix B.

The standard way to train a neural network is to use a gradient-based method. In the Two-Stage Predict+Optimize framework, we use the post-hoc regret $PReg$ as the loss function. Therefore, for each edge weight $w_e$ in the neural network, we need to compute the derivative $\frac{\mathrm{d}PReg}{\mathrm{d}w_e}$. Using the law of total derivative, we get

$$\frac{\mathrm{d}PReg(\hat{\theta}, \theta)}{\mathrm{d}w_e} = \frac{\partial PReg(\hat{\theta}, \theta)}{\partial x_2^*}\bigg|_{x_1^*} \frac{\partial x_2^*}{\partial x_1^*} \frac{\partial x_1^*}{\partial \hat{\theta}} \frac{\partial \hat{\theta}}{\partial w_e} + \frac{\partial PReg(\hat{\theta}, \theta)}{\partial x_1^*}\bigg|_{x_2^*} \frac{\partial x_1^*}{\partial \hat{\theta}} \frac{\partial \hat{\theta}}{\partial w_e} \tag{3}$$

As such, we wish to calculate each term on the right hand side.

The easiest term to handle is $\frac{\partial \hat{\theta}}{\partial w_e}$, since $\hat{\theta}$ is the neural network output, and so the derivatives can be directly calculated by standard backpropagation [25]. As for the terms $\frac{\partial PReg(\hat{\theta}, \theta)}{\partial x_2^*}\big|_{x_1^*}$ and $\frac{\partial PReg(\hat{\theta}, \theta)}{\partial x_1^*}\big|_{x_2^*}$, they are easily calculable whenever both the optimization objective and penalty function are smooth, and in fact linear as in the case of MILPs. What remains are the terms $\frac{\partial x_2^*}{\partial x_1^*}$ and

$\frac{\partial x_1^*}{\partial \hat{\theta}}$. The challenge is that $x_2^*$ is the solution of a MILP optimization (Stage 2) that uses $x_1^*$ as its parameters, i.e., differentiate through a MILP. Similarly, $x_1^*$ depends on $\hat{\theta}$ through a MILP (Stage 1). Since MILP optima may not change under minor parameter perturbations, the gradients can be either 0 or non-existent, which are uninformative. We thus need to compute some approximation in order to get useful training signals.

Our approach, inspired by the work of Mandi and Guns [18], is to define a new surrogate loss function $\widetilde{PReg}$ that is differentiable and produces informative gradients. Prior works related to learning unknowns in constraints [1, 2, 27] give ways of differentiating through LPs or LPs with regularizations. These works can be used in place of the proposed approach. However, experiments in Appendix E demonstrate that the proposed approach performs at least as well in post-hoc regret performance as the others, while being faster. We show the construction of the proposed approach below, and note that it does not have a simple closed form. Nonetheless, we can compute its gradients.

The rest of the section assumes that both stages of optimization are expressible as a MILP in the following standard form:

$$x^* = \arg\min_x c^\top x \text{ s.t. } Ax = b, Gx \geq h, x \geq 0, x_S \in \mathbb{Z} \tag{4}$$

with decision variables $x \in \mathbb{R}^d$ and problem parameters $c \in \mathbb{R}^d$, $A \in \mathbb{R}^{p \times d}$, $b \in \mathbb{R}^p$, $G \in \mathbb{R}^{q \times d}$, $h \in \mathbb{R}^q$. The subset of indices $S$ denotes the set of variables that are under integrality constraints. Since the unknown parameters may appear in any combination of $c, A, b, G$ and $h$ in the Stage 1 optimization for $x_1^*$, the surrogate loss function construction needs computable and informative gradients for all of $\frac{\partial x^*}{\partial c}$, $\frac{\partial x^*}{\partial A}$, $\frac{\partial x^*}{\partial b}$, $\frac{\partial x^*}{\partial G}$ and $\frac{\partial x^*}{\partial h}$.

We follow the interior-point based approach of Mandi and Guns [18], used also by Hu et al. [12]. Consider the following convex relaxation of (4), for a *fixed* value of $\mu \geq 0$:

$$x^* = \arg\min_{x,s} c^\top x - \mu \sum_{i=1}^{d} \ln(x_i) - \mu \sum_{i=1}^{q} \ln(s_i) \text{ s.t. } Ax = b, Gx - s = h \tag{5}$$

This is a relaxation of (4) by i) dropping all integrality constraints, ii) introducing slack variables $s \geq 0$ to turn $Gx \geq h$ into $Gx - s = h$ and iii) replacing both the $x \geq 0$ and $s \geq 0$ constraints with the logarithm barrier terms in the objective, with multiplier $\mu \geq 0$. The observation is that the gradients $\frac{\partial x}{\partial c}$, $\frac{\partial x}{\partial A}$, $\frac{\partial x}{\partial b}$, $\frac{\partial x}{\partial G}$ and $\frac{\partial x}{\partial h}$ for (5) are all well-defined, computable and informative for a *fixed* value of $\mu \geq 0$: Slater's condition holds for (5), and so the KKT conditions must be satisfied at the optimum $(x^*, s^*)$ of (5). We can thus compute all the relevant gradients via differentiating the KKT conditions, using the implicit function theorem. We give all the calculation details in Appendix B.

Given the above observation, we then aim to construct the surrogate loss function by replacing the $x_1^*$ and $x_2^*$, which are supposed to solved using MILP (4), with a) $\widetilde{x}_1$ that is solved from program (5) relaxation of the Stage 1 optimization problem, using the predicted parameters $\hat{\theta}$ and b) $\widetilde{x}_2$ that is solved from the program (5) relaxed version of Stage 2 optimization, using $\widetilde{x}_1$ and the true parameters $\theta$. The only remaining question then, is, which values of $\mu$ do we use for the two relaxed problems?

Given a MILP in the form of (4), the interior-point based solver of Mandi and Guns [18] generates and solves (5) for a sequence of decreasing non-negative $\mu$, with a termination condition that $\mu$ cannot be smaller than some cutoff value. Thus, we simply choose the cutoff value to use as "$\mu$" in (5), which then completes the definition of the surrogate loss $\widetilde{PReg}$.

Algorithmically, we train the neural network on the surrogate loss $\widetilde{PReg}$ as follows: given predicted parameters, we run the Mandi and Guns solver to get the optimal solution $(x^*, s^*)$ for the final value of $\mu$. We can then compute the gradient of the output solution with respect to any of the problem parameters using the calculations in Appendix B, combined with backpropagation, to yield $\frac{d\widetilde{PReg}}{dw_e}$ according to Equation (3).

In Appendix C, we give three example application scenarios, along with their penalty functions, that our training approach can handle. These problems are: a) an alloy production problem, for factory trying to source ores under uncertainty in chemical compositions in the raw materials, b) a variant of the classic 0-1 knapsack with unknown weights and rewards, and c) a nurse roster scheduling problem with unknown patient load. We show explicitly in Appendix C how both stages of optimization

Table 1: Relevant problem sizes of the three benchmarks.

| Problem name | Brass alloy production | Titanium-alloy production | 0-1 knapsack | Nurse scheduling problem |
|---|---|---|---|---|
| Dimension of x | 10 | 10 | 10 | 315 |
| Number of constraints | 12 | 14 | 21 | 846 |
| Number of unknown parameters | 20 | 40 | 10 | 21 |
| Number of features (per parameter) | 4096 | 4096 | 4096 | 8 |

can be formulated as MILPs for these applications, and apply the Appendix B calculations to yield gradient computation formulas for the surrogate loss $\widetilde{PReg}$ for these problems.

A limitation of our approach is the requirement that both stages must be expressible as MILPs, constraining the optimization objectives to be linear in the MILP decision variables. This contrasts the Hu et al. framework [12] which handles non-linear penalties. We point out that even MILPs can handle some non-linearity by using extra decision variables: for example, the absolute-value function. Moreover, the Appendix B gradient calculations can be adapted to handle general differentiable non-linear objectives. We present only MILPs as a main overarching application for this paper because of their widespread use in discrete optimization, with readily available solvers.

## 5    Experimental Evaluation

We evaluate the proposed method[2] on three benchmarks described in Section 4 and Appendix C. Table 1 reports the relevant benchmark problem sizes. We compare our method (2S) with the state of the art Predict+Optimize method, IntOpt-C [12], and $5$ classical regression methods [9]: ridge regression (Ridge), $k$-nearest neighbors ($k$-NN), classification and regression tree (CART), random forest (RF), and neural network (NN). All of these methods use their classic loss function to train the prediction models. At test time, to ensure the feasibility of the solutions when computing the post-hoc regret, we perform Stage 2 optimization on the estimated solutions for these classical regression methods before evaluating the final solution. Additionally, CombOptNet [23] is a different method focusing on learning unknowns in constraints, but with a different goal and loss function. We experimentally compare our proposed method with CombOptNet on the 0-1 knapsack benchmark—the only with available CombOptNet code. We also present a qualitative comparison in Section 6.

In the following experiments, we will need to take care to distinguish two-stage optimization as a training technique (Section 4) and as an evaluation framework (Section 3). We will denote our training method as "2S" in the experiments, and when we say "Two-Stage Predict+Optimize" framework, we always mean it as an evaluation framework. 2S is always evaluated according to the Two-Stage Predict+Optimize framework. As explained above, we will also evaluate all the classical training methods using the Two-Stage Predict+Optimize framework. For our comparison with the prior work of Hu et al. [12], we will also distinguish their training method and evaluation framework. The name "IntOpt-C" always refers to their training method using their correction function. We will simply call their evaluation framework the "Hu et al. framework" or with similar phrasing (see Section 2 to recall details). IntOpt-C will sometimes be evaluated using our new Two-Stage Predict+Optimize framework, and sometimes the prior framework of Hu et al. [12] using their correction function.

The methods of $k$-NN, RF, NN, and IntOpt-C as well as 2S have hyperparameters, which we tune via cross-validation. We include the hyperparameter types and chosen values in Appendix D. In the main paper we only report the prediction performances. See Appendix H for runtime comparisons.

**Alloy Production Problem**    The alloy production problem is a covering LP, see Appendix C.1 for the practical motivation and LP model. Since Hu et al. [12] also experimented on this problem, we use it to compare our 2S method with IntOpt-C [12], using the same dataset and experimental setting.

We conduct experiments on the production of two real alloys: brass and an alloy blend for strengthening Titanium. For brass, 2 kinds of metal materials, Cu and Zn, are required [14]. The blend of the two materials are, proportionally, $req = [627.54, 369.72]$. For the titanium-strengthening alloy, $4$ kinds of metal materials, C, Al, V, and Fe, are required [15]. The blend of the four materials are proportional to $req = [0.8, 60, 40, 2.5]$. We use the same real data as that used in IntOpt-C [12] as numerical values in our experiment instances. In this dataset [23], each unknown metal concentration

---

[2]Our implementation is available at https://github.com/Elizabethxyhu/NeurIPS_Two_Stage_Predict-Optimize

Table 2: Comparison of the Two-Stage Predict+Optimize framework and the Hu et al. framework on the alloy production problem.

| PReg | | Two-Stage Predict | Hu et al. |
|---|---|---|---|
| Alloy | Penalty factor | +Optimize Framework | Framework |
| Brass | 0.25±0.015 | **43.87±2.73** | 68.16±6.26 |
| | 0.5±0.015 | **65.71±4.81** | 82.91±5.45 |
| | 1±0.015 | **88.75±5.91** | 107.64±6.85 |
| | 2±0.015 | **123.90±6.84** | 150.47±12.99 |
| | 4±0.015 | **161.86±8.49** | 178.69±10.09 |
| | 8±0.015 | **194.06±13.09** | 206.84±12.51 |
| Titanium-alloy | 0.25±0.015 | **4.52±0.47** | 6.45±0.81 |
| | 0.5±0.015 | **6.03±0.62** | 7.90±0.56 |
| | 1±0.015 | **8.58±0.74** | 10.73±0.81 |
| | 2±0.015 | **12.17±1.24** | 14.17±1.31 |
| | 4±0.015 | **16.10±1.06** | 17.48±0.99 |
| | 8±0.015 | **19.69±0.91** | 21.08±1.91 |

Table 3: Mean post-hoc regrets and standard deviations for the alloy production problem using the Two-Stage Predict+Optimize framework.

| PReg | | 2S | IntOpt-C | Ridge | $k$-NN | CART | RF | NN | TOV |
|---|---|---|---|---|---|---|---|---|---|
| Alloy | Penalty factor | | | | | | | | |
| Brass | 0.25±0.015 | **43.87±2.73** | 45.27±3.35 | 60.80±2.55 | 63.32±4.39 | 77.80±6.37 | 60.85±2.35 | 64.96±3.58 | |
| | 0.5±0.015 | **65.71±4.81** | 67.69±4.25 | 71.12±3.48 | 74.36±5.69 | 93.67±7.03 | 70.86±3.29 | 74.32±2.90 | |
| | 1±0.015 | **88.75±5.91** | 89.83±4.79 | 91.82±6.41 | 96.52±8.90 | 125.50±9.49 | 90.97±6.14 | 93.12±4.24 | 312.02±6.94 |
| | 2±0.015 | **123.90±6.84** | 125.46±9.26 | 133.18±12.98 | 140.77±16.02 | 189.12±16.10 | 131.12±12.48 | 130.67±10.52 | |
| | 4±0.015 | **161.86±8.49** | 164.94±10.33 | 215.87±26.54 | 229.22±30.74 | 316.31±30.95 | 211.40±25.56 | 205.76±24.33 | |
| | 8±0.015 | **194.06±13.09** | 200.42±8.51 | 381.30±53.75 | 406.19±60.42 | 570.75±61.42 | 372.01±51.82 | 355.96±52.25 | |
| Titanium-alloy | 0.25±0.015 | **4.52±0.47** | 4.72±0.58 | 6.43±0.39 | 6.13±0.34 | 7.07±0.45 | 5.75±0.48 | 6.56±0.59 | |
| | 0.5±0.015 | **6.03±0.62** | 6.23±0.64 | 7.71±0.45 | 7.27±0.39 | 8.57±0.45 | 6.76±0.55 | 7.38±0.67 | |
| | 1±0.015 | **8.58±0.74** | 8.71±0.95 | 10.26±0.62 | 9.55±0.52 | 11.57±0.52 | 8.76±0.72 | 9.03±0.84 | 30.27±0.54 |
| | 2±0.015 | **12.17±1.24** | 12.31±1.31 | 15.37±1.03 | 14.11±0.84 | 17.57±0.80 | 12.78±1.11 | 12.34±1.21 | |
| | 4±0.015 | **16.10±1.06** | 16.97±1.70 | 25.60±1.89 | 23.24±1.56 | 29.57±1.53 | 20.81±1.93 | 18.95±2.00 | |
| | 8±0.015 | **19.69±0.91** | 20.80±1.74 | 46.04±3.65 | 41.49±3.03 | 53.57±3.10 | 36.88±3.63 | 32.16±3.60 | |

is related to 4096 features. For experiments on both alloys, 350 instances are used for training and 150 instances for testing the model performance. For NN, IntOpt-C, and 2S, we use a 5-layer fully connected network with 512 neurons per hidden layer.

In the penalty function described in Appendix C.1, we need to choose a penalty factor/multiplier for each supplier. We conduct experiments on 6 types of penalty factor ($\sigma$) settings: 6 vectors where each entry is i.i.d. uniformly sampled from $[0.25 \pm 0.015]$, $[0.5 \pm 0.015]$, $[1.0 \pm 0.015]$, $[2.0 \pm 0.015]$, $[4.0 \pm 0.015]$, and $[8.0 \pm 0.015]$ respectively. This random sampling of $\sigma$ ensures that the penalty factor for each supplier is different, but remains roughly on the same scale.

The first experiment we run compares 2S+Two-Stage Predict+Optimize framework with IntOpt-C+Hu et al. framework. Specifically, we compare a) using 2S for training and evaluating using the Two-Stage Predict+Optimize framework in Section 3, versus b) using IntOpt-C for training and evaluating using the same correction function from training, according to the Hu et al. framework described in Section 2. Table 2 compares the mean post-hoc regret and standard deviations for the alloy production problem for the two different frameworks. The table shows that Two-Stage Predict+Optimize framework always achieves smaller mean post-hoc regret than the Hu et al. framework. Compared with the Hu et al. framework, our framework obtains 6.18%-35.63% smaller mean post-hoc regret in brass production, and 6.59%-29.89% smaller mean post-hoc regret in titanium-alloy production.

We present a further comparison in Appendix F with a variant of the Hu et al. framework—the $\ell_2$ projection idea in [3], which performs even worse than the Hu et al. framework.

The second experiment compares various training approaches *all* evaluated under the Two-Stage Predict+Optimize framework. That is, the models are trained differently, but at test time, we always use Stage 2 optimization to give a final solution and evaluate post-hoc regret on it. Table 3 reports the mean post-hoc regrets and standard deviations across 10 runs for each training method on the alloy production problem. The table shows that our method, 2S, achieves the best performance, compared with IntOpt-C achieving the second best performance, beating all the classical training approaches. Compared with IntOpt-C, 2S obtains 1.20%-3.18% smaller mean post-hoc regrets in brass production, and 1.18%-5.33% smaller mean post-hoc regret in titanium-alloy production. Compared with the classical approaches, the improvements are much more significant. 2S obtains at least 2.44%-45.48% smaller mean post-hoc regrets in brass production, and at least 1.39%-38.78% smaller mean post-hoc regret in titanium-alloy production. The average True Optimal Values (TOV) are reported in the last column of Table 3 for reference, although the reader should take care to *not* over-interpret the ratio

Table 4: Mean post-hoc regrets and standard deviations for 0-1 knapsack problem using the Two-Stage Predict+Optimize framework.

| PReg | Penalty factor | 2S | CombOptNet | Ridge | $k$-NN | CART | RF | NN | TOV |
|---|---|---|---|---|---|---|---|---|---|
| 100 | 0.21 | **1.26±0.01** | 9.45±0.19 | 9.46±0.19 | 9.38±0.21 | 8.67±0.13 | 9.50±0.26 | 9.81±0.20 | 29.68±0.14 |
| | 0.25 | **6.28±0.05** | 9.60±0.22 | 9.77±0.19 | 9.70±0.19 | 9.19±0.12 | 9.82±0.27 | 10.11±0.20 | |
| | 0.3 | **9.22±0.10** | 10.45±0.34 | 10.16±0.19 | 10.10±0.18 | 9.85±0.11 | 10.22±0.28 | 10.49±0.21 | |
| 150 | 0.21 | **0.73±0.01** | 8.90±8.97 | 9.12±0.22 | 8.91±0.20 | 8.46±0.18 | 9.20±0.27 | 9.66±0.47 | 40.23±0.19 |
| | 0.25 | **3.64±0.04** | 9.11±9.41 | 9.40±0.21 | 9.19±0.20 | 8.88±0.17 | 9.47±0.26 | 9.92±0.43 | |
| | 0.3 | **7.27±0.06** | 9.34±9.38 | 9.76±0.22 | 9.53±0.19 | 9.41±0.17 | 9.81±0.24 | 10.23±0.38 | |
| 200 | 0.21 | **0.33±0.01** | 15.16±0.21 | 6.57±0.21 | 6.38±0.29 | 6.26±0.21 | 6.59±0.23 | 7.08±0.95 | 48.13±0.24 |
| | 0.25 | **1.67±0.03** | 15.20±0.27 | 6.80±0.20 | 6.62±0.29 | 6.57±0.19 | 6.82±0.21 | 7.27±0.88 | |
| | 0.3 | **3.33±0.06** | 15.25±0.22 | 7.09±0.19 | 6.91±0.28 | 6.95±0.19 | 7.10±0.18 | 7.52±0.80 | |
| 250 | 0.21 | **0.07±0.00** | 20.42±0.25 | 2.39±0.22 | 2.18±0.20 | 2.45±0.20 | 2.34±0.32 | 2.70±1.34 | 53.43±0.26 |
| | 0.25 | **0.34±0.02** | 20.47±0.13 | 2.53±0.21 | 2.34±0.19 | 2.60±0.19 | 2.49±0.30 | 2.82±1.26 | |
| | 0.3 | **0.69±0.04** | 20.54±0.32 | 2.71±0.20 | 2.54±0.18 | 2.79±0.18 | 2.67±0.28 | 2.97±1.16 | |

Table 5: Mean post-hoc regrets and standard deviations for the NSP using the Two-Stage Predict+Optimize framework.

| Penalty factor | 2S | Ridge | $k$-NN | CART | RF | NN | TOV |
|---|---|---|---|---|---|---|---|
| 0.25±0.015 | **3.94±1.91** | 6.45±4.68 | 15.20±5.76 | 26.20±8.96 | 19.47±7.19 | 4.27±2.22 | |
| 0.5±0.015 | **6.92±2.26** | 12.68±9.35 | 30.29±11.53 | 52.47±17.96 | 38.93±14.42 | 8.20±4.40 | |
| 1.0±0.015 | **13.12±3.15** | 25.12±18.71 | 60.43±23.07 | 105.01±36.00 | 77.86±28.99 | 16.00±8.78 | 190.21±26.17 |
| 2.0±0.015 | **25.04±9.29** | 49.95±37.39 | 120.62±46.08 | 210.02±72.06 | 155.64±58.06 | 31.51±17.40 | |
| 4.0±0.015 | **33.29±9.53** | 99.61±74.78 | 241.01±92.14 | 420.04±144.18 | 311.19±116.23 | 62.52±34.64 | |
| 8.0±0.015 | **46.72±14.80** | 198.91±149.54 | 481.79±184.27 | 840.10±288.45 | 622.32±232.56 | 124.54±69.14 | |

between the post-hoc regret and the true optimal value, since the post-hoc regret also includes the penalty term which increases with the penalty factors.

**0-1 knapsack** In the second example, we showcase our framework on a packing integer programming problem, a variant of the 0-1 knapsack problem, with unknown item prices $p_i$ and sizes $s_i$. See Appendix C.2 for details of an application in running a "proxy buyer" business. Here, the unknown parameters appear in both the objective and constraints. The proposed 2S method can handle this MILP straightforwardly, but the IntOpt-C method cannot be applied. Thus, we only experiment with the Two-Stage Predict+Optimize framework for evaluation, and compare the proposed 2S method with classical approaches and CombOptNet. Again, all approaches are evaluated at test time using the Stage 2 optimization to yield the final solution, on which the post-hoc regret is computed.

The MILP formulation of the two stages and the penalty function are described in Appendix C.2. We use the dataset of Paulus et al. [23], in which each 0-1 knapsack instance consists of 10 items and each item has 4096 features related to its price and size. For both NN and our method, we use a 5-layer fully-connected network with 512 neurons per hidden layer. We conduct experiments on 4 different knapsack capacities: 100, 150, 200, and 250. We use 700 instances for training and 300 instances for testing the model performance. Considering the real-life setting, we use 3 scales of the penalty factor for the penalty function in Appendix C.2: $\sigma = 0.05, 0.25$, or $0.5$.

Table 4 reports the mean post-hoc regrets and standard deviations across 10 runs for each approach on this 0-1 knapsack problem. Due to the space limitation and the fact that larger penalty factors are unrealistic in this problem setting, we present penalty factors $\geq 1$ in Appendix G. The average True Optimal Values (TOV) are reported in the last column, again for reference. As shown in the table, our proposed 2S method has significantly better results. In addition, we observe that across all approaches, the post-hoc regrets decrease as the knapsack capacity increases: this is due to the fact that as the capacity increases, more and more items can be selected, and so minor inaccuracies in predicted values/weights do not affect the selected set of items as much. On the other hand, the advantage of our 2S method over other approaches actually becomes more significant as the capacity increases, demonstrating the superior accuracy of our approach.

**Nurse Scheduling Problem** Our last experiment is on the nurse scheduling problem (NSP) with unknown patients needs, with the goal of scheduling a nurse roster satisfying unknown patient load demands while minimizing mismatched nurse-shift preferences as the objective. See Appendix C.3 for a description of the application scenario, the MILP formulations of the two stages, as well as the associated penalty function. Given that NSP is not an LP, IntOpt-C again does not apply, and so we

only compare the proposed 2S training method with the classical approaches, using the Two-Stage Predict+Optimize framework for evaluation. Each NSP instance consists of 15 nurses, 7 days, and 3 shifts per day. The nurse preferences are obtained from the NSPLib dataset [26], which is widely used for NSP [16, 20]. The number of patients that each nurse can serve in one shift is randomly generated from [10,20], representing the fact that each nurse has different capabilities. Given that we are unable to find datasets specifically for the patient load demands and relevant prediction features, we follow the experimental approach of Demirovic et al. [4, 5, 6] and use real data from a different problem (the ICON scheduling competition) as the numerical values required for our experiment instances. In this dataset, the unknown number of patients per shift is predicted by 8 features.

Since there are far fewer features than the previous experiments, for both NN and 2S we use a smaller network structure: a 4-layer fully-connected network with 16 neurons per hidden layer. We use 210 instances for training and 90 instances for testing. Just like the first experiment, we use 6 scales of penalty factors (see Appendix C.3 for the penalty function): $\gamma$ with i.i.d. entries drawn uniformly from $[0.25 \pm 0.015]$, $[0.5 \pm 0.015]$, $[1.0 \pm 0.015]$, $[2.0 \pm 0.015]$, $[4.0 \pm 0.015]$, and $[8.0 \pm 0.015]$.

Table 5 reports the mean post-hoc regrets and standard deviations across 10 runs for each approach on the NSP. The table shows that the proposed 2S method again has the best performance among all the training approaches. Our 2S method obtains at least 7.61%, 15.65%, 17.99%, 20.51%, 46.76%, and 62.49% smaller post-hoc regret than other classical methods when the penalty factor is $[0.25 \pm 0.015]$, $[0.5 \pm 0.015]$, $[1.0 \pm 0.015]$, $[2.0 \pm 0.015]$, $[4.0 \pm 0.015]$, and $[8.0 \pm 0.015]$ respectively.

**Runtime Analysis** Appendix H gives the training times for each method. Most classical approaches are faster than our 2S method, although as shown their post-hoc regrets are much worse. In alloy production, the only setting where IntOpt-C applies, its running time is shorter but comparable with 2S. In 0-1 knapsack, the only problem with public CombOptNet code, the 2S method is much faster.

## 6 Literature Review

Section 1 already summarized prior works in Predict+Optimize, most of which focus on learning unknowns only in the objective. Only the Hu et al. [12] framework considers unknowns in constraints.

Here we summarize other works related to learning unknowns in optimization problem constraints, particularly those outside of Predict+Optimize. These works can be placed into two categories.

One category also considers learning unknowns in constraints, but with very different goals and measures of loss. For example, CombOptNet [23] and Nandwani et al. [21] focus on learning parameters so as to make the predicted optimal solution (first-stage solution in our proposed framework) as close to the true optimal solution $x^*$ as possible in the solution space/metric. By contrast, our proposed framework explicitly formulates the two-stage framework and post-hoc regret in order to directly capture rewards and costs in application scenarios. Experiments on 0-1 knapsack in Section 5 show that these other methods yield worse predictive performance when evaluated on the post-hoc regret, under the proposed two-stage framework.

Another category gives ways to differentiate through LPs or LPs with regularizations, as a technical component in a gradient-based training algorithm. As mentioned in Section 4, these works can indeed be used in place of our proposed approach in Section 4/Appendix B. However, we point out that: (i) these other technical tools are essentially orthogonal to our primary contribution, which is the two-stage framework (Section 3), and (ii) nonetheless, experiments on the 0-1 knapsack in Appendix E demonstrate that our gradient calculation approach performs at least as well in post-hoc regret performance as other works, while being faster.

## 7 Summary

We proposed Two-Stage Predict+Optimize: a new, conceptually simpler and more powerful framework for the Predict+Optimize setting where unknown parameters can appear both in the objective and in constraints. We showed how the simpler perspective offered by the framework allows us to give a general training framework for all MILPs, contrasting prior work which apply only to covering and packing LPs. Experimental results demonstrate that our training method offers significantly better prediction performance over other classical and state-of-the-art approaches.

## Acknowledgments

We thank the anonymous referees for their constructive comments. In addition, Xinyi Hu and Jimmy H.M. Lee acknowledge the financial support of a General Research Fund (RGC Ref. No. CUHK 14206321) by the University Grants Committee, Hong Kong. Jasper C.H. Lee was supported in part by the generous funding of a Croucher Fellowship for Postdoctoral Research, NSF award DMS-2023239, NSF Medium Award CCF-2107079 and NSF AiTF Award CCF-2006206.

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

# A Detailed discussion on the Two-Stage Predict+Optimize framework

## A.1 Problem modelling using the framework

As mentioned in Section 3, the proposed Two-Stage Prediction+Optimize framework is phrased differently from some other two-stage problem formulations. The proposed framework phrases Stage 1 solutions as *soft* commitments, and corrects Stage 1 solutions with *penalty* in Stage 2. On the other hand, some two-stage frameworks phrase Stage 1 solutions as *hard* commitments, and include explicit *recourse variables* in both stages of OP to model the correction in Stage 2. Some optimization problems are more natural to express according to one perspective than the other, while some problems might be straightforward to express in either. This section aims to show that our framework, while explicitly stated and motivated according to the first perspective, is in fact general enough to also easily model the second perspective of hard commitments and recourse actions. In what follows, we first describe different types of variables and how our framework can capture them. Then, we give two example problems that respectively use the soft/hard commitment perspectives, and we detail how the problem can be modelled.

**Soft commitment variables:** These are variables which represent decisions that correspond to *soft* commitments made in Stage 1 in an application, namely decisions that may be modified once the true parameters are revealed, but at a cost or penalty. The discussion in Section 3 is tailored for this kind of variables—simply define such a variable in Stage 1 and use a finite penalty function to model the cost of changing this soft commitment in Stage 2.

**Hard commitment variables:** These are variables $x^*_{\text{hard}}$ which represent *hard* commitments made in Stage 1, meaning that after commitment, they absolutely cannot change in Stage 2. To model these variables in our framework, simply write a penalty function that is infinite whenever Stage 1 and Stage 2 solutions for these variables are different. Explicitly, add a term $\infty \cdot \mathbb{1}[x^*_{\text{hard},1} \neq x^*_{\text{hard},2}]$. This way, no Stage 2 solution will change these variables from what they were committed to in Stage 1.

**Recourse/other variables:** These are variables which represent explicit actions/decisions taken only in Stage 2, once the true parameters are revealed. These variables are necessary, for example, when Stage 1 actions are all hard commitment variables, to ensure that we have a mechanism for corrective action if the hard Stage 1 decisions are in any way "incompatible" with the revealed parameters. These corrective actions also typically come at a cost. Thus, to model these variables, simply include them in both Stages 1 and 2, and incorporate their cost into the objective of the optimization problem. There should also be 0 penalty for modifying these variables between the stages.

To summarize, Stage 1 actions can be classified as either soft or hard commitments, depending on whether they can be changed in Stage 2 (at a finite penalty). Stage 2 actions are classified as "recourse" variables, which are simply variables that have no penalty from changing between Stage 1 to Stage 2. The above discussion shows how our framework captures all these possibilities. We now give two example applications: the first one is more naturally expressed via the soft commitment perspective, and the second one is more natural to phrase using hard commitments+recourse. We give also their explicit formulations to demonstrate how the modelling is done in our framework.

We first show an example problem which is naturally modelled using soft commitment variables and penalty functions. Consider the product-stocking problem in Example 1 again, where regular orders have to be placed two weeks ahead of monthly deliveries. We aim to maximize the net profit by selling stocked products, under the constraint that the available storage space is limited. Each product $i$ has a purchase price $p^u_i$ (the price of purchasing the product from the wholesale company) and a selling price $p^s_i$ (the price of selling the product to customers), and needs $s_i$ space to be stocked. Let $x_i$ denote whether the product $i$ is ordered. In Stage 1, i.e., two weeks before the delivery, the available storage space $Sp$ at the time of delivery is unknown, and we place the order $x$ based on estimated space. In Stage 2, i.e., the night before the delivery, the precise available space is revealed, and we ask the wholesale company to change the order but need to pay a surcharge for last-minute changes. Assume the surcharge for the last-minute change in the order of product $i$ is $c_i$. In this example, $x_i$ is thus a soft commitment variable, and we model the surcharge $c_i$ using the penalty function of the framework.

The proposed framework can naturally model this problem. The Stage 1 OP can be formulated as:

$$x_1^* = \arg\max_x \sum_i (p_i^s - p_i^u) x_i$$

$$\text{s.t.} \sum_i s_i x_i \leq \hat{S}p, \quad x \in \{0, 1\}$$

In Stage 2, the order $x_1^*$ can be changed with surcharges, which can be modelled as a penalty function:

$$Pen(x_1^* \to x) = \sum_i c_i |x_1^* - x_i|$$

Then the Stage 2 OP can be formulated as:

$$x_2^* = \arg\max_x \sum_i (p_i^s - p_i^u) x_i - \sum_i c_i |x_1^* - x_i|$$

$$\text{s.t.} \sum_i s_i x_i \leq Sp, \quad x \in \{0, 1\}$$

Next, we give an example problem which is more naturally modelled using hard commitment variables and recourse variables. Consider a production-planning problem: a company owns a set of facilities and provides services to a set of customers. Each facility $i$ can provide a fixed amount of services $m_i$ and has a fixed operating cost $f_i$ in the standard working mode. The company aims to meet customer demands $d$ at the minimum operating costs. In Stage 1, the company decides which facilities to open for production based on the estimated demands $\hat{d}$. This is a binary decision variable $x_i$ for each facility $i$. In Stage 2, the orders from customers arrive and the demands $d$ are revealed. If the services provided by the operating facilities in the standard mode cannot meet demands, the company will ask some facilities that are already operating (i.e. $x_i = 1$) to work overtime, but naturally need to pay high overtime fees. Let $o_i$ denote the unit overtime fee for producing service in facility $i$, and $\sigma_i$ denote the amount of service provided by overtime working in facility $i$.

This example is naturally modelled using hard commitment variables and recourse variables. Which facilities to operate, $x$, is a vector of 0/1 hard commitment variables. The amount of service, $\sigma$, provided by the overtime working mode of operating facilities can be modeled by recourse variables, and the recourse costs are the overtime fees $o$. Using hard commitment variables and recourse variables, the Stage 1 OP can be formulated as:

$$x_1^*, \sigma_1^* = \arg\min_{x,\sigma} \sum_i f_i x_i + \sum_i o_i \sigma_i$$

$$\text{s.t.} \sum_i (m_i + \sigma_i) x_i \geq \hat{d}, \quad x \in \{0, 1\}, \quad \sigma \geq 0$$

In Stage 2, we include a term $\infty \cdot \mathbb{1}[x_1^* \neq x]$ in the penalty function part of the Stage 2 objective to make sure that $x$ cannot be changed, while the penalty for changing $\sigma$ is zero since it is a recourse variable. The Stage 2 OP is formulated as:

$$x_2^*, \sigma_2^* = \arg\min_{x,\sigma} \sum_i f_i x_i + \sum_i o_i \sigma_i + \infty \cdot \mathbb{1}[x_1^* \neq x]$$

$$\text{s.t.} \sum_i (m_i + \sigma_i) x_i \geq d, \quad x \in \{0, 1\}, \quad \sigma \geq 0$$

In summary, we discussed how to model in our framework soft and hard commitment actions in Stage 1, as well as recourse/other actions in Stage 2. We gave two concrete examples to demonstrate how such modelling can be done.

## A.2 What if correction/recourse is not possible in the application?

The motivating premise of this paper is that the application scenario at hand allows for some post-hoc corrective action once the true parameters are revealed. One natural question is: what if such

corrective action (Stage 2 actions) is not actually possible in the application? For example, in our running example of the product-stocking problem, we considered a wholesale company that allows for order changes the night before. Other wholesalers may not allow such a correction/modification. Our framework can essentially still model these scenarios: just set the penalty of modification to infinity (or at least, very large numbers for practice). Concretely, use the penalty function $\infty \cdot \mathbb{1}[x_2^* \neq x_1^*]$ (or replace $\infty$ with a very large number). This penalty function encourages the learning algorithm to learn *conservative* predictions that maximize the chances of yielding Stage 1 decisions that remain feasible in Stage 2.

To show this, we ran another quick experiment, using the 0-1 knapsack problem setting in the paper (with knapsack capacity = 100). This time, as we varied the magnitude of the penalty function, we measure at test time the empirical fraction of Stage 1 solutions that remain feasible under the true parameters. The results in Table 6 demonstrate our claim that as the penalty term increases, the predictions get more and more likely to remain feasible, making it a reasonable way to train a predictor even when Stage 2 correction mechanisms do not actually exist in the application.

Table 6: Mean and standard deviation of empirical fraction of Stage 1 solutions that remain feasible in Stage 2, for the 0-1 knapsack problem when capacity is 100 using the Two-Stage Predict+Optimize framework.

| Penalty Factor | Feasibility% |
|---|---|
| 0.05 | 0.00%±0.00% |
| 0.25 | 0.00%±0.00% |
| 0.5 | 1.73%±0.52% |
| 1 | 50.93%±1.92% |
| 2 | 51.63%±1.22% |
| 4 | 99.07%±0.31% |

### A.3 Two-Stage Predict+Optimize vs Prior Hu et al. Framework

As mentioned earlier in Section 3, Two-Stage Predict+Optimize is technically mathematically equivalent to the prior framework of Hu et al. [12], in the sense of expressiveness, *ignoring* differentiability issues. On the one hand, we can regard the Stage 2 optimization as a form of correction function, and hence Two-Stage Predict+Optimize can be considered as a special case of the Hu et al. [12] framework. On the other hand, given a correction function as in the Hu et al. [12] framework, we can simply modify the penalty function such that we keep the penalty value of the corrected solution, and make the penalty value infinite for all other potential Stage 2 solutions. This forces the Stage 2 optimization to always emulate the correction function. In this sense, our Two-Stage Predict+Optimize framework can also emulate the Hu et al. [12] framework, meaning that the two frameworks are technically equivalent.

Nevertheless, the Two-Stage Predict+Optimize framework is both *conceptually simpler* and *easier to apply*. In the main paper, we showed how to perform end-to-end neural network training within this new framework whenever both stages of optimization can be phrased as MILPs, and also give empirical experimental results. Together, they demonstrate the much more general applicability of the Two-Stage Predict+Optimize framework.

We end this appendix with the statement and short proof that, conditioned on the same penalty function and prediction model, Two-Stage Predict+Optimize always outputs at least as good a final solution as the prior framework using any correction function.

**Proposition A.1.** *Consider an arbitrary* minimization *Para-OP $P$, penalty function $Pen$, correction function $x_{corr}^*$, estimated parameters $\hat{\theta}$ and true parameters $\theta$. Let $x^*(\hat{\theta})$ and $x_1^*(\hat{\theta})$ both denote the estimated solution from the estimated parameters $\hat{\theta}$, $x_2^*(\hat{\theta}, \theta)$ be the output final solution from the Two-Stage Predict+Optimize framework, and $x_{corr}^*(\hat{\theta}, \theta)$ be the output corrected solution from the prior framework of Hu et al. Then,*

$$obj(x_2^*(\hat{\theta}, \theta), \theta) + Pen(x_1^*(\hat{\theta}) \to x_2^*(\hat{\theta}, \theta)) \leq obj(x_{corr}^*(\hat{\theta}, \theta), \theta) + Pen(x^*(\hat{\theta}) \to x_{corr}^*(\hat{\theta}, \theta))$$

*Proof.* Observe that both sides of the inequality are the objective of the Stage 2 optimization problem, evaluated at $x_2^*$ and $x_{corr}^*$ respectively. Since $x_2^*$ is the optimal solution to the minimization problem, the inequality follows directly. $\square$

# B  Gradient Calculations for Problem (5)

**Approximating** $\frac{\partial x_1^*}{\partial \hat{\theta}}$. In the context of the MILP, the unknown parameter $\hat{\theta}$ may either be $c, A, b, G$, or $h$. Using the solution $x$ and the barrier weight $\mu$ returned from solving Problem (5), we can compute the relevant derivatives of $\frac{\partial x_1^*}{\partial \hat{c}}$. The case of $c$ has already been derived by Mandi and Guns [18] (see Appendix A.1 and A.2 in their paper). Problem (5) can be rewritten as:

$$x^* = \arg\min_{x'} c'^\top x' - \mu \sum_{i=1}^{d+q} \ln(x_i') \tag{6}$$

$$\text{s.t. } A'x' = b'$$

where

$$
\begin{aligned}
c' &= \begin{bmatrix} c & 0 \end{bmatrix} \in \mathbb{R}^{d+q} \\
x' &= \begin{bmatrix} x & s \end{bmatrix} \in \mathbb{R}^{d+q} \\
A' &= \begin{bmatrix} A & 0 \\ G & -I \end{bmatrix} \in \mathbb{R}^{(p+q)\times(d+q)} \\
b' &= \begin{bmatrix} b \\ h \end{bmatrix} \in \mathbb{R}^{p+q}
\end{aligned}
$$

**Fact B.1.** *Consider the LP relaxation (6), defining $x'$ as a function of $c'$, $A'$ and $b'$. Then, according to Mandi and Guns [18], under this definition of $x^*$,*

$$
\begin{bmatrix} -X'^{-1}T & A'^\top & -c' \\ A' & 0 & -b' \\ -c'^\top & b'^\top & \frac{\kappa}{\tau} \end{bmatrix}
\begin{bmatrix} \frac{\partial x'}{\partial c'} \\ \frac{\partial y'}{\partial c'} \\ \frac{\partial \tau}{\partial c'} \end{bmatrix}
=
\begin{bmatrix} \tau I \\ 0 \\ x^\top \end{bmatrix}
$$

*where $X' = diag(x'), t = \mu X'^{-1}e, T = diag(t)$, $y'$ is the lagrangian multiplier of Problem (6), and $\kappa$ and $\tau$ are additional variables added by Mandi and Guns [18] to represent the duality gap. The gradient $\frac{\partial x_1^*}{\partial \hat{c}}$ can be obtained by solving this system of equalities.*

Define the notation $f(x, c, G, h) = c^\top x - \mu \sum_{i=1}^{d} \ln(x_i) - \mu \sum_{i=1}^{q} \ln(G_i^\top x - h_i)$. Then, Problem (5) can be expressed as finding:

$$x^* = \arg\min_x f(x, c, G, h) \text{ s.t. } Ax = b \tag{7}$$

Using this notation, we write down the following four lemmas on computing $\frac{\partial x^*}{\partial G}$, $\frac{\partial x^*}{\partial h}$, $\frac{\partial x^*}{\partial A}$, and $\frac{\partial x^*}{\partial b}$ approximately.

**Lemma B.2.** *Consider the LP relaxation (7), defining $x^*$ as a function of $c, A, b, G$ and $h$. Then, under this definition of $x^*$,*

$$\frac{\partial x^*}{\partial G} = (H^{-1}A^\top(AH^{-1}A^\top)^{-1}AH^{-1} - H^{-1})f_{Gx}(x, c, G, h)$$

*where $H = f_{xx}(x, c, G, h)$ denotes the matrix of second derivatives of $f$ with respect to different coordinates of $x$, and similarly for other subscripts, and explicitly:*

$$f_{x_k x_j}(x, c, G, h) = \begin{cases} \mu x_j^{-2} + \mu \sum_{i=1}^{q} G_{ij}^2/(G_i^\top x - h_i)^2, & j = k \\ \mu \sum_{i=1}^{q} G_{ij}G_{ik}/(G_i^\top x - h_i)^2, & j \neq k \end{cases} \tag{8}$$

*and*

$$f_{G_{\ell r} x_j}(x, c, G, h) = \begin{cases} \mu G_{\ell j} x_j/(G_\ell^\top x - h_\ell)^2 - \mu/(G_\ell^\top x - h_\ell) & r = j \\ \mu G_{\ell j} x_r/(G_\ell^\top x - h_\ell)^2 & r \neq j \end{cases}$$

*Note that when there are no equality constraints, i.e., $A = 0$, we have*

$$\frac{\partial x^*}{\partial G} = -H^{-1}f_{Gx}(x, c, G, h)$$

*which is the same as the Lemma 3 in [12].*

*Proof.* Using the Lagrangian multiplier $y$, the Lagrangian relaxation of Problem (7) can be written as

$$\mathbb{L}(x, y; c, G, h) = f(x, c, G, h) + y^\top (b - Ax) \tag{9}$$

Since $x^* = \arg\min_x f(x, c, G, h)$ s.t. $Ax = b$ is an optimum, $x^*$ must obey the Karush-Kuhn-Tucker (KKT) conditions, obtained by setting the partial derivative of Equation (9) with respect to $x$ and $y$ to 0. Let $f_x(x, c, G, h)$ denotes the vector of first derivatives of $f$ with respect to different coordinates of $x$, $f_{xx}(x, c, G, h)$ denotes the matrix of second derivatives of $f$ with respect to different coordinates of $x$, we obtain:

$$\begin{aligned} f_x(x, c, G, h) - A^\top y &= 0 \\ Ax - b &= 0 \end{aligned} \tag{10}$$

The implicit differentiation of these KKT conditions with respect to $G$ allows us to get the following system of equalities:

$$\begin{bmatrix} f_{Gx}(x, c, G, h) \\ 0 \end{bmatrix} + \begin{bmatrix} f_{xx}(x, c, G, h) & -A^\top \\ A & 0 \end{bmatrix} \begin{bmatrix} \frac{\partial x}{\partial G} \\ \frac{\partial y}{\partial G} \end{bmatrix} = 0 \tag{11}$$

By solving this system of equalities, we can obtain

$$\frac{\partial x^*}{\partial G} = (H^{-1} A^\top (A H^{-1} A^\top)^{-1} A H^{-1} - H^{-1}) f_{Gx}(x, c, G, h)$$

Since $f(x, c, G, h) = c^\top x - \mu \sum_{i=1}^d \ln(x_i) - \mu \sum_{i=1}^q \ln(G_i^\top x - h_i)$, we have

$$\begin{aligned} f_{x_j}(x, c, G, h) &= c_j - \mu x_j^{-1} - \mu \sum_{i=1}^q G_{ij} / (G_i^\top x - h_i) \\ f_{x_k x_j}(x, c, G, h) &= \begin{cases} \mu x_j^{-2} + \mu \sum_{i=1}^q G_{ij}^2 / (G_i^\top x - h_i)^2, & j = k \\ \mu \sum_{i=1}^q G_{ij} G_{ik} / (G_i^\top x - h_i)^2, & j \neq k \end{cases} \end{aligned} \tag{12}$$

and

$$f_{G_{\ell r} x_j}(x, c, G, h) = \begin{cases} \mu G_{\ell j} x_j / (G_\ell^\top x - h_\ell)^2 - \mu / (G_\ell^\top x - h_\ell) & r = j \\ \mu G_{\ell j} x_r / (G_\ell^\top x - h_\ell)^2 & r \neq j \end{cases}$$

$\square$

**Lemma B.3.** *Consider the LP relaxation (7), defining $x^*$ as a function of $c, A, b, G$ and $h$. Then, under this definition of $x^*$,*

$$\frac{\partial x^*}{\partial h} = (H^{-1} A^\top (A H^{-1} A^\top)^{-1} A H^{-1} - H^{-1}) f_{hx}(x, c, G, h)$$

*where $H = f_{xx}$ is defined as in Lemma B.2 and*

$$f_{h_\ell x_j}(x, c, G, h) = -\mu G_{\ell j} / (G_\ell^\top x - h_\ell)^2$$

*Note that when there are no equality constraints, i.e., $A = 0$, we have*

$$\frac{\partial x^*}{\partial h} = -H^{-1} f_{hx}(x, c, G, h)$$

*which is the same as the Lemma 2 in [12].*

*Proof.* As stated in the proof of Lemma B.2, using the Lagrangian relaxation and the Karush-Kuhn-Tucker (KKT) conditions, we obtain:

$$\begin{aligned} f_x(x, c, G, h) - A^\top y &= 0 \\ Ax - b &= 0 \end{aligned} \tag{13}$$

The implicit differentiation of these KKT conditions with respect to $h$ allows us to get the following system of equalities:

$$\begin{bmatrix} f_{hx}(x, c, G, h) \\ 0 \end{bmatrix} + \begin{bmatrix} f_{xx}(x, c, G, h) & -A^\top \\ A & 0 \end{bmatrix} \begin{bmatrix} \frac{\partial x}{\partial h} \\ \frac{\partial y}{\partial h} \end{bmatrix} = 0 \tag{14}$$

By solving this system of equalities, we can obtain

$$\frac{\partial x^*}{\partial h} = (H^{-1} A^\top (A H^{-1} A^\top)^{-1} A H^{-1} - H^{-1}) f_{hx}(x, c, G, h)$$

where $H = f_{xx}$ is defined as in Lemma B.2. Since $f(x, c, G, h) = c^\top x - \mu \sum_{i=1}^{d} \ln(x_i) - \mu \sum_{i=1}^{q} \ln(G_i x - h_i)$, we have

$$f_{h_\ell x_j}(x) = -\mu G_{\ell j}/(G_\ell^\top x - h_\ell)^2$$

$\square$

**Lemma B.4.** *Consider the LP relaxation (7), defining $x^*$ as a function of $c, A, b, G$ and $h$. Then, under this definition of $x^*$,*

$$\frac{\partial x^*}{\partial A_{ij}} = H^{-1}(-A^\top(AH^{-1}A^\top)^{-1}(I_2 x + AH^{-1}I_1 y) + I_1 y)$$

*where $I_1 = -\frac{\partial A^\top}{\partial A_{ij}}, I_2 = \frac{\partial A}{\partial A_{ij}},$ and $H = f_{xx}$ is defined as in Lemma B.2.*

*Proof.* As stated in the proof of Lemma B.2, using the Lagrangian relaxation and the Karush-Kuhn-Tucker (KKT) conditions, we obtain:

$$\begin{aligned} f_x(x, c, G, h) - A^\top y &= 0 \\ Ax - b &= 0 \end{aligned} \tag{15}$$

Since $A \in \mathbb{R}^{p \times d}$, fix $i \in \{1, \ldots, p\}, j \in \{1, \ldots, d\}$, the implicit differentiation of these KKT conditions with respect to $A_{ij}$ allows us to get the following system of equalities:

$$\begin{bmatrix} -\frac{\partial A^\top}{\partial A_{ij}} y \\ \frac{\partial A}{\partial A_{ij}} x \end{bmatrix} + \begin{bmatrix} f_{xx}(x, c, G, h) & -A^\top \\ A & 0 \end{bmatrix} \begin{bmatrix} \frac{\partial x}{\partial A_{ij}} \\ \frac{\partial y}{\partial A_{ij}} \end{bmatrix} = 0 \tag{16}$$

Let $I_1 = -\frac{\partial A^\top}{\partial A_{ij}}, I_2 = \frac{\partial A}{\partial A_{ij}}$. By solving this system of equalities, we can obtain

$$\frac{\partial x^*}{\partial A_{ij}} = H^{-1}(-A^\top(AH^{-1}A^\top)^{-1}(I_2 x + AH^{-1}I_1 y) + I_1 y)$$

where $H = f_{xx}$ is defined as in Lemma B.2.

$\square$

**Lemma B.5.** *Consider the LP relaxation (7), defining $x^*$ as a function of $c, A, b, G$ and $h$. Then, under this definition of $x^*$,*

$$\frac{\partial x^*}{\partial b} = H^{-1}A^\top(AH^{-1}A^\top)^{-1}I$$

*where $H = f_{xx}$ is defined as in Lemma B.2.*

*Proof.* As stated in the proof of Lemma B.2, using the Lagrangian relaxation and the Karush-Kuhn-Tucker (KKT) conditions, we obtain:

$$\begin{aligned} f_x(x, c, G, h) - A^\top y &= 0 \\ Ax - b &= 0 \end{aligned} \tag{17}$$

The implicit differentiation of these KKT conditions with respect to $b$ allows us to get the following system of equalities:

$$\begin{bmatrix} 0 \\ -I \end{bmatrix} + \begin{bmatrix} f_{xx}(x, c, G, h) & -A^\top \\ A & 0 \end{bmatrix} \begin{bmatrix} \frac{\partial x}{\partial b} \\ \frac{\partial y}{\partial b} \end{bmatrix} = 0 \tag{18}$$

By solving this system of equalities, we can obtain

$$\frac{\partial x^*}{\partial b} = H^{-1}A^\top(AH^{-1}A^\top)^{-1}I$$

where $H = f_{xx}$ is defined as in Lemma B.2.

$\square$

# C  Details for Case Studies

Since the penalty function partly or solely affects the terms $\left.\frac{\partial PReg(\hat{\theta},\theta)}{\partial x_2^*}\right|_{x_1^*}$, $\left.\frac{\partial PReg(\hat{\theta},\theta)}{\partial x_1^*}\right|_{x_2^*}$, and $\frac{\partial x_2^*}{\partial x_1^*}$, we give three case studies for our framework to show how to design the penalty function and compute gradients using the corresponding penalty function.

## C.1  Alloy Production Problem

We first demonstrate, using the example of the alloy production problem, how our framework can tackle problems solvable by the prior work of Hu et al. [12]. An alloy production factory needs to produce a certain amount of a particular alloy, requiring a mixture of $M$ kinds of metals. To that end, it must acquire at least $req_m$ tons of each of the $m \in [M]$ metals. The raw materials are to be obtained from $K$ suppliers, each supplying a different type of ore. The factory plans to buy ores from sites and then extract the metals themselves. The ore supplied by site $k \in [K]$ contains a $con_{km} \in [0, 1]$ fraction of material $m$ at a price of $cost_k$ per ton. The goal of the factory is to meet its requirements for each metal at the minimum cost. However, the precise metal concentrations (averaged in a batch) are unknown before the factory actually completes metal extraction. The factory will estimate metal concentrations based on historical buying records, considering features such as the ore type, ore origin, site-reported preliminary samples and so on. Then the factory will decide how much ore to order from each site. This is the Stage 1 solution. The Stage 1 OP is the alloy production problem using the estimated metal concentrations $c\hat{o}n$, and can be formulated as follows:

$$x_1^* = \arg\min_x cost^\top x \qquad \text{s.t. } c\hat{o}n^\top x \geq req, \ x \geq 0$$

After the factory obtains the ores and completes metal extraction, i.e., in Stage 2, the precise metal concentrations/amounts are known. Since the purchased ores are already processed, the factory cannot return ores even if it has bought too much. However, if the obtained metals do not satisfy the requirements, the factory can post-hoc decide to last-minute order more ores at a higher price, for example, $(1 + \sigma_k)cost_k$ per ton from the site $k$, where $\sigma_k \geq 0$ is a non-negative tunable scalar parameter. In this scenario, the penalty function is:

$$Pen(x_1^* \to x) = (\sigma \circ cost)^\top (x - x_1^*) \tag{19}$$

where $\circ$ is the Hadamard/entrywise product.

With respect to the above penalty function, we are now ready to define the Stage 2 OP:

$$x_2^* = \arg\min_x cost^\top x + (\sigma \circ cost)^\top (x - x_1^*) \qquad \text{s.t. } con^\top x \geq req, \ x \geq x_1^* \tag{20}$$

Note that since the precise metal concentrations $con$ are revealed, the true concentrations are used as the problem parameters instead of the estimated concentrations. The final amount of ores bought from each site, including the ores bought in both Stage 1 and Stage 2, is the Stage 2 solution.

The above formulation is based on the "soft commitment" modelling approach discussed in Appendix A.1.

The post-hoc regret for the alloy production problem can be explicitly written as:

$$PReg(\hat{\theta}, \theta) = cost^\top x_2^* + (\sigma \circ cost)^\top (x_2^* - x_1^*) - cost^\top x^*(con) \tag{21}$$

where $x^*(con)$ is an optimal solution of the alloy production problem under the true concentrations $con$. We now show how to compute the relevant gradients as discussed in Section 4 and Appendix B.

Using Equation (21), it is straightforward to compute that the $i$-th item in vector $\left.\frac{\partial PReg(\hat{\theta},\theta)}{\partial x_2^*}\right|_{x_1^*}$ and the $i$-th item in vector $\left.\frac{\partial PReg(\hat{\theta},\theta)}{\partial x_1^*}\right|_{x_2^*}$: $\left(\left.\frac{\partial PReg(\hat{\theta},\theta)}{\partial x_2^*}\right|_{x_1^*}\right)_i = (1 + \sigma_i)\, cost_i$, $\left(\left.\frac{\partial PReg(\hat{\theta},\theta)}{\partial x_1^*}\right|_{x_2^*}\right)_i = -\sigma_i cost_i$.

Now we show how to compute the approximation of the remaining term, $\frac{\partial x_2^*}{\partial x_1^*}$.

**Approximation** $\frac{\partial x_2^*}{\partial x_1^*}$**.** We use the same interior-point LP solver to help compute the relevant derivatives. First, the estimated parameters are fed into the LP solver to solve the Stage 1 OP to obtain the Stage 1 optimal solution $x_1^*$ and the corresponding $\mu$, which are used to compute the term $\frac{\partial x_1^*}{\partial \theta}$. Then the Stage 1 optimal solution $x_1^*$ and the true parameters are fed into the LP solver to solve the Stage 2 OP to obtain the Stage 2 optimal solution $x_2^*$ and the corresponding $\mu$, which are used to compute the term $\frac{\partial x_2^*}{\partial x_1^*}$. Consider the Stage 2 OP in program (20). It is clear that the Stage 2 OP is a MILP, with $x_2^*$ in the objective and $x_1^*$ in $h$ of the constraints. Applying Lemma B.3, we can compute an approximate gradient of the $\frac{\partial x_2^*}{\partial x_1^*}$ term.

## C.2  Variant of 0-1 Knapsack

The second example, which we call the proxy buyer problem, is a variant of the 0-1 knapsack problem. The unknown parameters appear in both the objective and constraints. This problem, as we shall see, can be handled by our framework, but not by the prior approach by Hu et al. [12], since the problem is inherently discrete and cannot be formulated as LPs.

A *proxy buyer* is a person who purchases goods for others possibly for a profit. Consider a proxy buyer who is from City A, with a very high cost of living, who regularly travels to City B with a much lower cost of living. Given her regular travels, her friends in City A have asked her to help purchase everyday-life products, which are significantly cheaper in City B, yet the time and transportation cost from City A to City B makes it prohibitive for most people to just go to City B themselves. The traveller commutes between City A and City B once every three months, and has a known and limited capacity $cap$ of goods she could carry and bring back. Before each trip, her friends would make requests for things to buy. For simplicity, one request contains one item. If the buyer brings back the item as requested, her friends will pay her 20% of the price-tag $p_i$ of each item $i$ as a courtesy-thankyou. We denote this "profit" by $f_i$, i.e., $f_i = 20\% p_i$.

The buyer is popular, and many friends ask her for favours. One day before the buyer leaves for City B, the buyer needs to decide which of her friends' requests to accept, given the limited capacity, and inform them accordingly. The buyer wants to maximize the total amount of courtesy-thankyou money she gets, subject to the hard constraint of the limited suitcase capacity $cap$. However, the precise price $p_i$ of each item $i$ is unknown, due to the uncertainty of the price itself, the volatility of the exchange rate, and the uncertainty of the discount activities of the items. Thus, the "profit" $f_i$ of buying item $i$ is unknown. In addition, the exact size $s_i$ of each item $i$ is also estimated. The buyer will estimate the profit, i.e., the prices, and the sizes based on past experiences, considering features such as time-of-year, holiday-or-not, brand and so on. The buyer will decide which requests to accept based on the estimation. This is the Stage 1 solution. The Stage 1 OP is the proxy buyer problem using the estimated sizes $\hat{s}$ and estimated profits $\hat{f}$:

$$x_1^* = \arg\max_x \hat{f}^\top x, \qquad \text{s.t. } \hat{s}^\top x \leq cap, \ x \in \{0, 1\}$$

After the buyer arrives at City B, the buyer knows the precise price and size of each item. If she cannot carry all the accepted requests, for example, if the packaging for certain items have changed since she last bought them, the buyer will necessarily need to drop some of these requests. The buyer usually feels bad about reneging on a promise to her friends, and treats her friends to a meal as an apology if the request cannot be fulfilled after she promised. For simplicity, we assume that the price of the apology-meal is linear in the profit of the dropped request, since more expensive items are considered "more important" requests. Here, the linearity factor is independent of the request. That is, if she drops item $i$, she has to spend $\sigma f_i$ amount of money, where $\sigma \geq 0$ is a non-negative tunable scalar parameter. In this scenario, the penalty function is:

$$Pen(x_1^* \to x) = \sigma f^\top (x_1^* - x) \tag{22}$$

We are now ready to define the Stage 2 OP with respect to the above penalty function:

$$x_2^* = \arg\max_x f^\top x - \sigma f^\top (x_1^* - x), \qquad \text{s.t. } s^\top x \leq cap, \ x \leq x_1^*, \ x \in \{0, 1\} \tag{23}$$

The requests that were finally filled, namely the items that were actually bought by the buyer and brought home to City A, forms the Stage 2 solution.

Then the simplified form of the post-hoc regret for the proxy buyer problem can be written as:

$$PReg(\hat{\theta}, \theta) = f^\top x^*(f, s) - f^\top x_2^* + \sigma f^\top (x_1^* - x_2^*) \tag{24}$$

where $x^*(f, s)$ is an optimal solution of the proxy buyer problem under the true proxy fees $f$ and true sizes $s$.

Using Equation (24), it is straightforward to compute that the $i$-th item in vector $\left.\frac{\partial PReg(\hat{\theta}, \theta)}{\partial x_2^*}\right|_{x_1^*}$ and

the $i$-th item in vector $\left.\frac{\partial PReg(\hat{\theta}, \theta)}{\partial x_1^*}\right|_{x_2^*}$ : $\left(\left.\frac{\partial PReg(\hat{\theta}, \theta)}{\partial x_2^*}\right|_{x_1^*}\right)_i = (-1 - \sigma) f_i$, $\left(\left.\frac{\partial PReg(\hat{\theta}, \theta)}{\partial x_1^*}\right|_{x_2^*}\right)_i = \sigma f_i$.

**Approximation $\frac{\partial x_2^*}{\partial x_1^*}$.** Similar to the computation in Section C.1, we obtain the Stage 1 optimal solution $x_1^*$, the Stage 2 optimal solution $x_2^*$, and the corresponding $\mu$ from the interior-point LP compute the term $\frac{\partial x_2^*}{\partial x_1^*}$. Consider the Stage 2 OP in program (23), it is clear that the Stage 2 OP is a MILP, with $x_2^*$ in the objective and $x_1^*$ in $h$ of the constraints. Applying Lemma B.3, we can compute an approximate gradient of the $\frac{\partial x_2^*}{\partial x_1^*}$ term.

### C.3  Nurse Scheduling Problem

Our last example is the nurse scheduling problem (NSP), which can be handled by our framework but not by the prior work of Hu et al. [12] since it is neither a packing LP nor a covering LP.

Consider a large optometry center that needs to assign nurses to shifts per day to meet patients' needs. Every Monday morning, the center collects the nurses' preferences for each shift of the following week. Since nurses may have their own activities and errands during unscheduled shifts, they want to be informed of their schedules as early as possible. After the preferences are collected, on Monday night, the center sets a preliminary shift schedule for the upcoming week based on the estimated number of patients for each shift. Suppose there are $n$ nurses, $k$ days, and $s$ shifts per day, then the number of the total shifts is $t = k \times s$. We formulate the decision variables as a Boolean vector $x \in \{0, 1\}^d$, where $d = n \times k \times s$. Let $P \in \{1, 2, 3, 4\}^d$ represent the value of each nurse's preferences for a particular shift (the higher the number the better), and $H \in \mathbb{N}^t$ represents the number of patients in each shift, which are unknown and need to be predicted. Each nurse $i$ can serve $m_i$ patients in one shift. The objective is to maximize the nurses' preferences under a set of constraints: (1) the schedule must satisfy the patient demand, under each shift (2) each nurse must be scheduled for exactly one shift each day (3) no nurse may be scheduled to work a night shift followed immediately by a morning shift. The Stage 1 OP is the NSP using the estimated number of patients $\hat{H}$:

$$x_1^* = \arg\max_x P^\top x$$

$$\text{s.t.} \sum_{i=0}^{n-1} m_i x_{it+j} \geq \hat{H}_j \quad \forall j \in \{0, ..., t-1\}$$

$$\sum_{q=0}^{s-1} x_{it+sj+q} = 1 \quad \begin{aligned} &\forall i = \{0, \ldots, n-1\}, \\ &j = \{0, \ldots, k-1\} \end{aligned}$$

$$x_{it+sj+s-1} + x_{it+sj+s} \leq 1 \quad \begin{aligned} &\forall i = \{0, \ldots, n-1\}, \\ &j = \{0, \ldots, k-2\} \end{aligned}$$

$$x \in \{0, 1\}$$

To provide better service to patients, the optometry center has implemented an appointment system that requires patients to schedule an appointment in advance to receive medical care. Reservations for the upcoming week, from Monday to Sunday, close every Sunday evening. At this point, the center knows the precise number of patients for each shift of the next week. The center might adjust the shift schedule to satisfy the actual patient demand or to improve the overall nurse preferences. However, due to the late notice for schedule changes, the nurse's preference may become lower. For example, if a nurse is rescheduled to a shift for which her original preference is 5, now her preference for this shift may become 4 due to the late notice. Besides, a nurse may be more unhappy to be changed to

a low-preference shift. In this scenario, since the nurses' preferences are in $\{1, 2, 3, 4\}$, the penalty function can be formulated as:

$$Pen(x_1^* \to x) = \sum_{i=0}^{d-1} Pen(x_1^* \to x)_i \tag{25}$$

where the $i$-th item in the penalty function is:

$$Pen(x_1^* \to x)_i = \begin{cases} \gamma_i(5 - P_i)^2(x_i - x_{1i}^*) & x_i \geq x_{1i}^* \\ 0 & x_i < x_{1i}^* \end{cases}$$

We are now ready to define the Stage 2 OP with respect to the above penalty function:

$$x_2^* = \arg\max_x P^\top x - \sum_{i=0}^{d-1} Pen(x_1^* \to x)_i$$

$$\text{s.t.} \sum_{i=0}^{n-1} m_i x_{it+j} \geq H_j \quad \forall j \in \{0, ..., t-1\}$$

$$\sum_{q=0}^{s-1} x_{it+sj+q} = 1 \quad \begin{matrix} \forall i = \{0, \ldots, n-1\}, \\ j = \{0, \ldots, k-1\} \end{matrix}$$

$$x_{it+sj+s-1} + x_{it+sj+s} \leq 1 \quad \begin{matrix} \forall i = \{0, \ldots, n-1\}, \\ j = \{0, \ldots, k-2\} \end{matrix}$$

$$x \in \{0, 1\}$$

Then the simplified form of the post-hoc regret for the NSP can be written as:

$$PReg(\hat{\theta}, \theta) = P^\top x^*(H) - P^\top x_2^* + \sum_{i=0}^{d-1} Pen(x_1^* \to x_2^*)_i \tag{26}$$

Using Equation (26), it is straightforward to compute that the $i$-th item in vector $\left. \frac{\partial PReg(\hat{\theta}, \theta)}{\partial x_2^*} \right|_{x_1^*}$ and the $i$-th item in vector $\left. \frac{\partial PReg(\hat{\theta}, \theta)}{\partial x_1^*} \right|_{x_2^*}$:

$$\left( \left. \frac{\partial PReg(\hat{\theta}, \theta)}{\partial x_2^*} \right|_{x_1^*} \right)_i = \begin{cases} -P_i + 2\gamma_i(5 - P_i) & x_{2i}^* \geq x_{1i}^* \\ -P_i & x_{2i}^* < x_{1i}^* \end{cases}$$

$$\left( \left. \frac{\partial PReg(\hat{\theta}, \theta)}{\partial x_1^*} \right|_{x_2^*} \right)_i = \begin{cases} -2\gamma_i(5 - P_i) & x_{2i}^* \geq x_{1i}^* \\ 0 & x_{2i}^* < x_{1i}^* \end{cases}$$

**Approximation** $\frac{\partial x_2^*}{\partial x_1^*}$. Similar to the computation in Section C.1, we obtain the Stage 1 optimal solution $x_1^*$, the Stage 2 optimal solution $x_2^*$, and the corresponding $\mu$ from the interior-point LP compute the term $\frac{\partial x_2^*}{\partial x_1^*}$. Using the penalty function in Equation (25), the Stage 2 OP can be formulated

as a MILP by adding new variables $\sigma$ and one more constraint:

$$x_2^* = \arg\max_x P^\top x - \sum_{i=0}^{d-1} \gamma_i (5 - P_i)^2 \sigma_i$$

$$\text{s.t. } \sum_{i=0}^{n-1} m_i x_{it+j} \geq H_j \quad \forall j \in \{0, ..., t-1\}$$

$$\sum_{q=0}^{s-1} x_{it+sj+q} = 1 \quad \begin{aligned} \forall i &= \{0, \dots, n-1\}, \\ j &= \{0, \dots, k-1\} \end{aligned}$$

$$x_{it+sj+s-1} + x_{it+sj+s} \leq 1 \quad \begin{aligned} \forall i &= \{0, \dots, n-1\}, \\ j &= \{0, \dots, k-2\} \end{aligned}$$

$$\sigma_i \geq x_i - x_{1i}^* \quad \forall i = \{0, \dots, d-1\}$$

$$x \in \{0, 1\}$$

$$\sigma \in \{0, 1\}$$

Suppose the Stage 2 OP of the NSP can be written as:

$$x_2^* = \arg\min_x -P^\top x + (\gamma \circ (5 - P)^2)^\top \sigma$$

$$\text{s.t. } G_1 x \geq H$$

$$Ax = b$$

$$G_2 x \geq -1$$

$$\sigma - x \geq -x_1^*$$

$$x, \sigma \in \{0, 1\}$$

Then the standard form of the Stage 2 OP is:

$$x_2' = \arg\min_{x'} c^\top x'$$

$$\text{s.t. } A'x' = b$$

$$Gx \geq h$$

$$x' \in \{0, 1\}$$

where
$$c = \begin{bmatrix} -P & \gamma \circ (5 - P)^2 \end{bmatrix} \in \mathbb{R}^{2d}, \quad x' = \begin{bmatrix} x & \sigma \end{bmatrix} \in \mathbb{R}^{2d}$$
$$G = \begin{bmatrix} G_1 & 0 \\ G_2 & 0 \\ -I & I \end{bmatrix} \in \mathbb{R}^{(t+nk-n+d)\times 2d}, \quad h = \begin{bmatrix} H \\ -1 \\ -x_1^* \end{bmatrix} \in \mathbb{R}^{t+nk-n+d}$$
$$A' = \begin{bmatrix} A & 0 \end{bmatrix} \in \mathbb{R}^{nk\times 2d}$$

and $b \in \mathbb{R}^{nk}$ is an all-ones vector.

It is clear that $x_2^*$ is in the objective and $x_1^*$ is in $h$ of the constraints. Applying Lemma B.3, we can compute an approximate gradient of the $\frac{\partial x_2^*}{\partial x_1^*}$ term.

# D   Hyperparameters for the Experiments

The methods of $k$-NN, RF, NN, and IntOpt-C as well as 2S have hyperparameters, which we tune via cross-validation: for $k$-NN, we try $k \in \{1, 3, 5\}$; for RF, we try different numbers of trees in the forest $\{10, 50, 100\}$; for NN, IntOpt-C, and 2S, we treat the learning rate, epochs and weight decay as hyperparameters.

Tables 7, 8, and 9 show the final hyperparameter choices for the three problems: 1) an alloy production problem, 2) the classic 0-1 knapsack problem, and 3) a nurse roster scheduling problem.

Table 7: Hyperparameters of the experiments on the alloy production problem.

| Model | Hyperparameters |
|---|---|
| Proposed | optimizer: optim.Adam; learning rate: $5 \times 10^{-7}$; $\mu = 10^{-3}$; epochs=20 |
| $k$-NN | k=5 |
| RF | n_estimator=100 |
| NN | optimizer: optim.Adam; learning rate: $10^{-3}$; epochs=20 |

Table 8: Hyperparameters of the experiments on the 0-1 knapsack problem.

| Model | Hyperparameters |
|---|---|
| Proposed | optimizer: optim.Adam; learning rate: $10^{-7}$; $\mu = 10^{-3}$; epochs=12 |
| $k$-NN | k=5 |
| RF | n_estimator=100 |
| NN | optimizer: optim.Adam; learning rate: $10^{-3}$; epochs=12 |

Table 9: Hyperparameters of the experiments on the nurse scheduling problem.

| Model | Hyperparameters |
|---|---|
| Proposed | optimizer: optim.Adam; learning rate: $10^{-1}$; $\mu = 10^{-3}$; epochs=8 |
| $k$-NN | k=5 |
| RF | n_estimator=100 |
| NN | optimizer: optim.Adam; learning rate: $10^{-2}$; epochs=8 |

Ridge, $k$-NN, CART and RF are implemented using *scikit-learn* [24]. The neural network is implemented using *PyTorch* [22]. To compute the two stages of optimization at *test time* for our method, and to compute the optimal solution of an (MI)LP under the true parameters, we use the MILP solver from *Gurobi* [11].

# E   Comparisons of the 2S Method and the Prior Differentiation Methods

In this section, we compare the proposed method with prior works [1, 2, 27] that provide ways of differentiating through LPs or LPs with regularization. We conduct comparisons with CvxpyLayer [1] but not OptNet [2] or QPTL [27]. The reason is that the calculation method proposed in QPTL is LP+quadratic regularization using OptNet, and CvxpyLayer is just a conic extension to OptNet. We compared CvxpyLayer [1] with a) no regularization, b) quadratic regularization and c) log-barrier (like our Section 4/Appendix B). The key indicator of its predictive performance is the type of regularization used, with the log-barrier version performing the best, but still slightly worse than our method. We applied CvxpyLayer [1] to the 0-1 knapsack benchmark to compare with our 2S method.

Table 10 reports the mean post-hoc regrets and standard deviations across 10 runs and Table 11 reports the average training times. More precisely, we use it with various regularizations (a. LP with no regularization, b. with quadratic regularization, c. with log-barrier as in our paper) to replace the Section 4/Appendix B gradient calculations. We find that CvxpyLayer [1] never gives better solution quality while 2S is 30%–50% faster.

Table 10: Mean post-hoc regrets and standard deviations of the 2S method and CvxpyLayer with different regularization on the 0-1 knapsack problem.

| PReg | Penalty factor | 2S | CvxpyLayer+log | CvxpyLayer+quad_reg | CvxpyLayer+no_reg |
|---|---|---|---|---|---|
| cap=100 | 0.05 | **1.26±0.01** | **1.26±0.01** | **1.27±0.01** | 7.70±0.39 |
| | 0.25 | **6.28±0.05** | **6.28±0.05** | 6.34±0.03 | 8.87±0.92 |
| | 0.5 | **9.22±0.10** | 9.47±0.31 | 9.96±0.54 | 10.13±0.46 |
| cap=150 | 0.05 | **0.73±0.01** | **0.74±0.01** | **0.75±0.03** | 6.74±0.58 |
| | 0.25 | **3.64±0.04** | **3.64±0.04** | 3.70±0.03 | 7.18±0.77 |
| | 0.5 | **7.27±0.06** | **7.28±0.08** | 7.39±0.06 | 8.43±0.58 |

Table 11: Average runtime (in seconds) of the 2S method and CvxpyLayer with different regularization on the 0-1 knapsack problem.

| Runtime | 2S | CvxpyLayer+log | CvxpyLayer+quad_reg | CvxpyLayer+no_reg |
|---|---|---|---|---|
| cap=100 | 204.76 | 438.24 | 571.38 | 344.50 |
| cap=150 | 245.61 | 467.65 | 662.30 | 366.83 |

# F   Frameworks Comparisons on the Alloy Production Problem

In this section, we further compared the proposed framework with the framework using the differentiable projection idea in [3] on the alloy production benchmark. The idea in [3] is to use the $l_2$ projection, and we implemented it using CvxpyLayer. The experiment set-up follows that of Table 2: both training and testing use $l_2$ projection in the second stage, as opposed to solving the second stage optimization problem defined in Section 3. Table 12 shows both the post-hoc regret and training time for $l_2$ projection. We find that not only is $l_2$ projection slow, but it has even worse post-hoc regret than the Hu et al. correction [12]. We suspect that this is due to the Hu et al. correction function [12] preserving the direction of the solution vector whereas $l_2$ projection can change the direction, and that this makes a difference for Alloy Production. In any case, this experiment confirms again that our Two-Stage framework has better post-hoc regret than a framework based on differentiable projections, reinforcing the main message of our paper.

Table 12: Comparison of three frameworks on the alloy production problem.

| | PReg | | | | | | Average runtime |
|---|---|---|---|---|---|---|---|
| Penalty factor | 0.25±0.015 | 0.5±0.015 | 1±0.015 | 2±0.015 | 4±0.015 | 8±0.015 | |
| Two-Stage Predict + Optimize Framework | **43.87±2.73** | **65.71±4.81** | **88.75±5.91** | **123.90±6.84** | **161.86±8.49** | **194.06±13.09** | 268.22 |
| Hu et al. Framework | 68.16±6.26 | 82.91±5.45 | 107.64±6.85 | 150.47±12.99 | 178.69±10.09 | 206.84±12.51 | 228.00 |
| l2_projection | 103.28±4.87 | 118.90±6.99 | 150.15±11.45 | 212.62±20.58 | 337.59±23.24 | 562.41±34.29 | 442.97 |

# G   Experiments on the 0-1 Knapsack Problem with Large Penalty Factors

Table 13 reports the mean post-hoc regrets and standard deviations across 10 runs for each approach on the 0-1 knapsack problem with large penalty factors (penalty factors $\geq 1$). With more data, we can make further analysis of the performance of the proposed 2S method. Observing Tables 4 and 13, we can see that the trend, in terms of the difference between 2S and other methods, first decreases, then increases, as the penalty factor increases. The trend in Tables 4 and 13 is identical to the trend in Table 3. We can explain this phenomenon as follows.

First, when the penalty factor is small, the rational behavior for the buyer is to just take every order, and only decide which orders to drop when the true parameters are revealed (at close to no cost). 2S identifies and exploits this behavior for small penalties, while classic regression methods are agnostic to this possible tactic. Thus, the advantage of 2S compared to classic regression methods is large in the small penalty case.

Second, when the penalty factor is large, 2S will analogously learn to be conservative, such that the first stage solution likely remains feasible under the true parameters, in order to avoid the necessary (and high) penalty due to having to change to a feasible solution. Again, classic regression methods will be agnostic to this possible tactic, leading to a large advantage of 2S over the classic methods.

Table 5 only has the increasing trend from the large penalty, since it is neither a covering nor a packing program, and so there is no analogous tactic/exploitation for the small penalty.

Table 13: Mean post-hoc regrets and standard deviations for 0-1 knapsack problem with large penalty factors using the Two-Stage Predict+Optimize framework.

| PReg | Penalty factor | 2S | CombOptNet | Ridge | $k$-NN | CART | RF | NN | TOV |
|---|---|---|---|---|---|---|---|---|---|
| cap=100 | 1 | **10.90±0.15** | 10.93±0.17 | 10.93±0.19 | 11.11±0.17 | 11.16±0.14 | 11.01±0.31 | 11.26±0.23 | |
| | 2 | **12.31±0.16** | 12.45±0.25 | 12.48±0.20 | 12.49±0.21 | 13.77±0.26 | 12.60±0.39 | 12.78±0.30 | 29.68±0.14 |
| | 4 | **14.54±0.15** | 15.66±0.47 | 15.57±0.25 | 15.68±0.39 | 19.01±0.56 | 15.77±0.62 | 15.84±0.50 | |
| cap=150 | 1 | **10.23±0.12** | 10.22±0.18 | 10.46±0.23 | 10.40±0.18 | 10.46±0.19 | 10.49±0.21 | 10.86±0.30 | |
| | 2 | **11.18±0.15** | 11.74±0.34 | 11.88±0.30 | 11.63±0.20 | 12.56±0.31 | 11.83±0.19 | 12.12±0.17 | 40.23±0.19 |
| | 4 | **13.20±0.16** | 14.33±0.46 | 14.71±0.49 | 14.43±0.33 | 16.75±0.63 | 14.53±0.29 | 14.65±0.41 | |
| cap=200 | 1 | **6.77±0.36** | 15.30±0.28 | 7.67±0.18 | 7.51±0.27 | 7.71±0.20 | 7.67±0.16 | 8.00±0.65 | |
| | 2 | **8.19±0.12** | 15.39±0.16 | 8.84±0.22 | 8.69±0.26 | 9.24±0.30 | 8.80±0.20 | 8.97±0.37 | 48.13±0.24 |
| | 4 | **9.71±0.35** | 15.46±0.22 | 11.17±0.40 | 11.06±0.32 | 12.29±0.59 | 11.05±0.46 | 10.91±0.53 | |
| cap=250 | 1 | **1.37±0.08** | 20.69±0.20 | 3.08±0.19 | 2.94±0.16 | 3.17±0.17 | 3.05±0.25 | 3.28±0.96 | |
| | 2 | **3.34±0.15** | 20.78±0.20 | 3.80±0.20 | 3.73±0.15 | 3.94±0.20 | 3.79±0.26 | 3.89±0.58 | 53.43±0.26 |
| | 4 | **4.46±0.09** | 20.93±0.20 | 5.25±0.35 | 5.32±0.27 | 5.47±0.35 | 5.29±0.48 | 5.11±0.39 | |

# H  Runtimes for the Experiments

In this paper, all models are trained with Intel(R) Xeon(R) CPU E5-2630 v2 @ 2.60GHz processors. Table 14 shows the average runtime across 10 simulations for different optimization problems. Since the testing time of different approaches is quite similar, here, the runtime refers to only the training time of the prediction model and does not include the testing time. At training time, only the proposed 2S method and IntOpt-C solve the LP. Training for the usual NN does not involve the LP at all, and so training is much faster (but gives worse results).

Since IntOpt-C cannot handle the variant of the 0-1 knapsack problem and the NSP, we only report the runtime of IntOpt-C for the alloy production problem.

Since the provided code of CombOptNet is only available for the 0-1 knapsack problem, we only report the runtime of CombOptNet for the 0-1 knapsack problem. As Table 14 shows, CombOptNet is drastically slower than the proposed 2S method.

In the alloy production problem, the runtimes of the proposed 2S method are a little larger than that of IntOpt-C. The reason is that 2S needs to solve two LPs when training while IntOpt-C only needs to solve one. But in the alloy production problem, the unknown parameters are on the left hand side of the inequality constraints and the gradient computation includes matrix computation, which is also time-consuming. Thus, the runtimes of the 2S method are larger but not twice as large as that of the IntOpt-C method.

In both the alloy production problem and the variant of the 0-1 knapsack problem, the runtimes of the 2S method are much better than RF.

The runtime of the 2S method is large in the NSP. This is because we use the formulation where each decision variable corresponds to whether a specific nurse is assigned to a specific day and a specific shift. Thus, the number of the decision variable of the relaxed LP is large and the LP takes more time to solve.

Table 14: Average runtime (in seconds) for the alloy production, 0-1 knapsack, and nurse scheduling problems.

| Runtime(s) | Alloy production | | 0-1 knapsack | | | | Nurse scheduling |
|---|---|---|---|---|---|---|---|
| | Brass | Titanium-alloy | Capacity=100 | Capacity=150 | Capacity=200 | Capacity=250 | |
| 2S | 268.22 | 394.53 | 204.76 | 245.61 | 202.65 | 193.46 | 537.32 |
| IntOpt-C | 228.00 | 331.38 | N\A | | | | |
| CompOptNet | N\A | | 2341.40 | 2940.26 | 2394.05 | 2383.39 | N\A |
| Ridge | 20.22 | 56.89 | 22.33 | | | | <1 |
| $k$-NN | 25.14 | 70.22 | 26.00 | | | | <1 |
| CART | 30.33 | 94.89 | 34.83 | | | | <1 |
| RF | 959.50 | 2552.25 | 1034.07 | | | | 2.11 |
| NN | 212.22 | 321.11 | 135.80 | | | | 11.39 |

