# OpenReview forum: "Two-Stage Predict+Optimize for MILPs with Unknown Parameters in Constraints"
_NeurIPS.cc/2023/Conference — NeurIPS 2023 poster_

### Official Review · Reviewer_Jj2a · 2023-07-03

**Soundness:** 2 fair
**Presentation:** 3 good
**Contribution:** 3 good
**Rating:** 7
**Confidence:** 4

**Summary:**

Predict+Optimize is an emerging paradigm that lies in the intersection of classical optimization (particularly mixed integer programming) and machine learning. Specifically, it considers the setting where a parameterized optimization problem:
$$ x^{\star}(\theta) = \operatorname*{argmin}_{x} f(x;\theta) \text{ subject to } C(x;\theta) $$
must be solved yet the parameters $\theta$ are unknown. Here $C(x;\theta)$ represents constraints that must be respected, which could be equality, inequality, or inclusion. In the last 4--5 years this topic has seen heightened interest. Most approaches, including the one proposed in this paper, assume access to observed features $A$ that correlate with $\theta$, and aim to learn a predictor $f(A)$ so that $\hat{\theta} := f(A) \approx \theta$. Then, a problem of the form above is solved with $\hat{\theta}$ in place of $\theta$ to yield a predicted minimizer $x^{\star}(\hat{\theta})$. Hopefully,  $x^{\star}(\hat{\theta}) \approx x^{\star}(\theta)$.


The paper in question is one of the first to actively consider the setting where the constraints depend on $\theta$; prior work mostly considered constraints of the form $C(x)$. The method proposed is interesting, intuitive and general. They start from the observation that any scheme handling parametrized constraints must be allowed to make post-hoc adjustments to $x^{\star}(\hat{\theta})$  after the true parameters $\theta$ are revealed, as $x^{\star}(\hat{\theta})$ might not be feasible. Thus, they propose to solve a second, penalized problem:

$$ x_2^{\star} = \operatorname{argmin}_x f(x;\theta) + r(x^{\star}(\hat{\theta}), x) \text{ subject to } C(x;\theta) $$

where $r(\bullet, \bullet)$ penalizes the discrepancy between the stage 1 solution $x^{\star}(\hat{\theta})$ and the stage 2 solution $x_2^{\star}$.

The bulk of the paper is devoted to motivating, introducing and implementing this proposed new method. Numerical experiments illustrate that this new method performs well. Finally, there is a little bit of theory (contained in the appendix) relating this new approach to older approaches to predict+optimize with parametrized constraint sets.




**Strengths:**

- I agree with the assertion in the appendix that the proposed approach is _simple and powerful_. I think this is an elegant solution to an important problem.
- The paper is well-written; I did not find any typos or unclear sentences. The running example of stocking a store is useful.
- I appreciated the choice of experiments---they are closer to "real-world" problems than many experiments I have encountered in Neurips papers.
- The Appendices are thoughtfully written, addressing many potential reader's questions and providing several nice extensions of the model at hand.
- The framework proposed is indeed original, but I think some additional comparisons with existing literature is needed, see "weaknesses".




**Weaknesses:**

- With regard to novelty, I think the authors should compare their work to various differentiable quadratic program (QP) solvers. For example, in [1] formulas for the derivatives of $x^{\star}$ with respect to constraint (_e.g._ $\partial x^{\star}/\partial A$, to reference the notation of your equation (5)) are given. Note that the MILP in (4) can be turned into a QP by using a quadratic regularizer (as done in [2]) instead of the logarithmic regularizer of Mandi & Guns.

 - I found the use of the name "Two-stage Predict+Optimize" for your proposed framework a bit confusing. In much of the Predict+Optimize literature, the two-stage approach refers to the approach of training a predictor $f(A)$ to minimize the mean square error $\|f(A) - \theta\|^2$ (see for example [3]) I strongly suggest you add a remark  discussing this towards the beginning of your paper.

- I think the benchmarks considered are a little weak. For example, all 5 classical regression methods are slight variations on the "two-stage approach" mentioned above. You should include some SOTA approaches for (one-stage) predict+optimize, e.g. training with the SPO+ loss, Perturbed Optimization [7], blackbox backpropagation [8] etc or justify why such approaches are incompatible with the proposed Two-Stage Predict+Optimize framework. The PyEPO benchmarking suite [4] could be useful.

- I'd like to see more discussion on the computational cost of your proposed approach. Given that it is a tri-level (!) problem, I think this is something to address. For the experiments, could you indicate the dimension of $x$ and the number of constraints, as well as the time required to train using the 2S and IntOpt-C approaches? It might be nice to add a remark about possible approaches to scaling your proposed framework, see [5,6].

[1] _Optnet: Differentiable optimization as a layer in neural networks_ Amos & Kolter, 2017.

[2] _Melding the data-decisions pipeline: Decision focused learning for combinatorial optimization_ Wilder _et al_, 2019

[3] _Interior point solving for LP-based prediction+optimization_ Mandi & Guns, 2020

[4] _PyEPO: A PyTorch-based end-to-end predict-then-optimize library for linear and integer programming_ Tang & Khalil 2022

[5] _Faster predict-and-optimize using three-operator splitting_ McKenzie _et al_, 2023

[6] _Backpropagation through combinatorial solvers: Identity with projection works_, Sahoo _et al_, 2022

[7] _Learning with differentiable perturbed optimizers_ Berthet _et al_, 2020

[8] _Differentiation of blackbox combinatorial solvers_ Vlastelica _et al_, 2019



**Questions:**

- The two-stage nature of your proposed approach reminds me a lot of Model-agnostic Meta-Learning (MAML) [2]. Moreover, I think there is an interesting analogy between the connection between your work and that of Hu et al [3] and the connection between MAML and later implicit MAML (I-MAML) works [4]. I think the appeal of your work could be broadened by adding a discussion of this connection.

- An interesting variant of predict+optimize is the setting where at train time the true cost vectors ($c$ in the notation of your equation (4)) are never revealed, rather only the true solutions $x^{\star}$ are accessible (see [1] for a discussion on this). Do you have any thoughts on how to extend your approach to this setting?

[1]  _Faster predict-and-optimize using three-operator splitting_ McKenzie _et al_, 2023

[2] _Model-agnostic meta-learning for fast adaptation of deep networks_ Finn _et al_, 2017

[3] _Predict+Optimize for packing and covering LPs with unknown parameters in constraints_ Hu _et al_ 2022

[4] _Meta-learning with implicit gradients_ Rajeswaran _et al_, 2019.

-----

After discussion with the authors I've raised my score 6--> 7

-------

**Limitations:**

As mentioned above, I think the computational complexity of the proposed approach could be a limitation. This should be discussed.

---

> ### Author Rebuttal · Authors · 2023-08-10
>
> We appreciate your positive review of our paper, concrete suggestions and also your questions. We respond to your points below to address your remaining reservations about our work. Hopefully you are more convinced by our contributions. Please let us know if you have any additional questions.
>
> -----
>
> Response to "weaknesses":
>
> 1. We added comparisons to cvxpylayers using quadratic regularization (and no regularization and log-barrier regularization). Please see the overall response and our response to reviewer Xm8w.
>
> 2. We will clarify in the paper, thank you for pointing this out.
>
> 3. One-stage Predict+Optimize methods referenced in the review, by definition, cannot handle unknown parameters in constraints (since Hu et al. was the first Predict+Optimize framework for unknowns in constraints), and therefore they do not apply and are incompatible with our new Two-Stage Predict+Optimize framework. See also Table 1 in the PyEPO paper for reference. Moreover, we added additional comparisons with cvxpylayers.
>
> 4. For the experiments in the paper, see the following table for the number of decision variables (dimension of $x$), constraints, unknown parameters and features. Training times are reported in Appendix E, and as discussed in the overall response, 2S has rather comparable training time compared with IntOpt-C. In the rebuttal experiments, our training times are also faster than using cvxpylayers instantiated using various regularizations.
> We agree that it is an interesting future work direction to speed up training and further improve the training-time scalability; the main focus of our work is on getting good learning performance.
>
> | Problem name                 | Brass alloy production | Titanium-alloy production | 0-1 knapsack | Nurse scheduling problem |
> |------------------------------|:----------------------|:-------------------------|:--------------|:--------------------------|
> | Dimension of $x$        | 10           |             10            |      10      |            315           |
> | Number of constraints  |   12           |             14            |      21      |            846           |
> | Number of unknown parameters | 20           |             40            |      10      |            21            |
> | Number of features (per parameter) |4096          |            4096           |     4096     |             8            |
>
> ------
>
> **Q1**: Thank you for pointing out the high-level similarities between our framework and that of MAML. Here is a brief comparison between the two.
>
> *Similarities*: In both MAML and our setting, the prediction isn't evaluated/used directly. Instead, after prediction happens, new information is revealed by the environment that allows us to adapt the model. In our framework, this is the second stage optimization after the true parameters are revealed. In the meta-learning setting, it is the true task being revealed, and the model can be fine-tuned. Training needs to be aware of this adaptation, in order to perform well.
>
> *Differences*: The most important difference of course is that the adaptations are quite different: second stage optimization (our setting) vs fine-tuning (meta-learning). Also, in MAML, there are no features for the future task, and the future task is drawn purely distributionally (and we assume that the training algorithm has access to samples). On the other hand, in Two-Stage Predict+Optimize, we have features for predicting the true parameters. In this sense, MAML is closer to traditional stochastic programming, and our framework in a sense is a *contextual* variant. Though of course, one could plausibly also formulate a contextual version of MAML to bring it closer to our setting.
>
> **Q2**: This is a really interesting question, and we believe it is enough scope for another paper. Here we present our initial thoughts on the problem, with basic theory. It remains to empirically test whether this approach would actually work.
>
> We'll consider the same two-stage setting as in our paper, where we get features in the first stage, need to make a soft commitment using predictions, and then once the true parameters are revealed we solve for a second stage solution.
> The difference is that, now, for training, we get feature-solution pairs $(A, x^\ast)$, where $x^\ast$ is the optimal solution under the true parameters $\theta$, instead of feature-(true parameter) $(A,\theta)$ pairs.
>
> Assuming the penalty is non-negative and that the penalty for no solution modification is 0, we train a predictor $\hat{\theta}(A)$ so as to minimize
> $Pen(x_1 \to x^\ast,\hat{\theta}(A))$
> where $x_1 = \mathrm{argmin}\text{ } obj(x, \hat{\theta}(A))$ s.t. $C(x, \hat{\theta}(A))$ holds.
>
> This is somewhat similar to what McKenzie et al. already does, except that our first stage optimization also has unknowns in constraints, and that we use the problem-specific penalty function as opposed to a generic l2 loss.
>
> Note the following two basic lemmas justifying this training loss, which are straightforward to prove:
>
> 1. It is minimized at $\hat{\theta}(A)$ that induces $x_1 = x^\ast$, yielding a training loss of 0. So for example, $\hat{\theta}(A) = \theta$ is a minimizer.
>
> 2. Suppose further that the penalty function has no explicit dependence on the unknown parameters, and only depends on the first and second stage solutions. Then, for every $\hat{\theta}(A)$, we have $\text{training loss} + obj(x^\ast,\theta) = Pen(x_1 \to x^\ast) + obj(x^\ast, \theta) \ge Pen(x_1 \to x_2) + obj(x_2, \theta) = \text{test loss}$, where $x_2$ is the second stage solution induced by the predicted parameters $\hat{\theta}(A)$ and the true parameters $\theta$.
> Since $obj(x^\ast,\theta)$ does not depend on the predictions $\hat{\theta}(A)$, it does make sense to train to minimize the training loss, in order to minimize the test loss.
>
> We believe that this is a reasonable starting point for this different Predict+Optimize problem, but this is out of the scope of the current paper.

---

> > ### Comment · Reviewer_Jj2a · 2023-08-17
> >
> >
> > Thanks to the authors for their thorough rebuttal! After reading the other reviews, I have some additional questions. Then, I will respond to some of the points raised in your response
> >
> > --------
> >
> > 1. Reviewer Pw5Y points out that your method can only use linear penalty functions. I looked over the paper and couldn't find any discussion of what a suitable penalty function should be, besides from an oblique reference in line 223-224. Could you elaborate on what penalty functions are admissable and why?
> > 2. I am still a little confused by the benchmark approaches of Section. 5. If I understand correctly, you use ridge regression, k-NN etc to learn a prediction function $\hat{\theta} = f(A)$ by solving
> > $$ \min_{f}\sum_{i=1}^{n}\|f(A^i) - \theta^i\|_2^2 $$
> > (or similar loss function). Then, at test time you solve Stage 1 using $\hat{\theta}$, then Stage 2 using $\hat{\theta}, \theta$ to obtain $x^{\star}_2$. Then $x^{\star}_2$ is plugged into $\mathrm{PReg}(\theta, \hat{\theta})$ to obtain the values listed in Table 2 and Table 3. Is this correct?
> >
> > --------
> >
> > Response to response to "weaknesses"
> >
> > 1. Thanks for implementing additional benchmarks, particularly cvxpylayers. In your implementation, do you use it to solve both Stage 1 and Stage 2?
> > 3. You're right. Most of the standard P&O methods I listed can't handle parameters in the constraints, so you are justified in not including them as benchmarks.
> > 4. Thanks for providing this table! I highly recommend you include it in the final version.
> >
> > ------
> >
> > Thanks for humoring my two somewhat off-topic questions. Some further thoughts:
> >
> > **Q1:** In addition, the point I was trying to convey is that the evolution from MAML to iMAML was triggered by realizing that a one-step correction process (in MAML this is a single step of gradient descent) can be reframed as a secondary optimization problem. I think this is analogous to the evolution from the method of Hu et al to the proposed method.
> >
> > **Q2:** Again, thanks for exploring this connection! It is interesting, but I agree it is out of the scope of this paper.
> >
> > ------
> >
> > I am satisfied with the authors response. Assuming the paper is edited so as to include additional benchmarks and expand the literature, I would be happy to see it accepted at NeuRIPS. I will adjust my score accordingly.

---

> > > ### Author Response · Authors · 2023-08-18
> > >
> > > Thank you for your additional comments and questions. We will incorporate the discussion raised in the reviews and our rebuttal, including the additional benchmarks and literature review, into the final version.
> > >
> > > -----
> > >
> > > **Questions**
> > > 1. The design of the penalty function is guided by the real-world application and is thus a modeling issue. The precise characterization of the penalty functions in our Section 4 is: as long as the Stage 2 optimization (which includes the penalty function) is expressible as a MILP. This includes linear penalties, and also convex/concave piecewise linear penalties (for minimization/maximization resp.) through the introduction of auxiliary variables. Linear penalties are widely applicable in real-world scenarios and can cover many realistic problems.  Even if the penalty is indeed non-linear in nature, we can often use a linear approximation which is sufficient for most practical purposes.
> > > 2. Yes, your understanding is correct.
> > >
> > > -----
> > >
> > > **Response to response**
> > > 1. Yes, our implementation uses cvxpylayers in both Stages 1 and 2.
> > >
> > > -----
> > >
> > > **"Off-topic" questions**
> > >
> > > 1. Thank you for further pointing out the similarities to (i)MAML. We will digest this more fully and discuss in the final version.

---

### Official Review · Reviewer_Pw5Y · 2023-07-05

**Soundness:** 3 good
**Presentation:** 3 good
**Contribution:** 3 good
**Rating:** 7
**Confidence:** 4

**Summary:**

The paper presents a novel '2-stage' framework for Predict+Optimize with uncertain parameters in the constraints. In the first "stage" a soft commitment is made based on the predictions, and in the second, the commitment is updated based on updated information in such a way that the objective value plus a penalty for deviating from the commitment is minimized. This generalizes the framework of Hu et al. [9]

**Strengths:**

1. Great idea: I find the new framework simple and sensible, I think this is the right framework for Predict+Optimize with unknown constraints. The biggest plus is that it allows for improving soft commitments in which constraints aren't violated, which is not something that Hu et al. allow for, but makes a lot of practical sense. It also removes the need to create a differentiable projection.
1. Clarity: The paper is well written and covers most bases (the tables could be bigger, though!)

**Weaknesses:**

1. Requires linear penalties: The penalties have to be linear in the decision variables. Hu et al. [9] doesn't require that.
1. Is slower than Hu et al.: Because Hu et al. only require running a differentiable projection, which can be quite cheap, it is cheaper than 2S (Appendix E confirms this).
1. Generalization of Hu et al.'s framework: While Hu et al. propose a specific projection that is limited to packing and constraint problems, there has been work in the literature about differential projections for enforcing feasibility constraints generally [A] and efficiently [B] that could be used instead. It would have been great to compare against those in the experiments.
1. I understand that the space is limited, but a lot of important information, like the description of the experiments and runtimes are in the Appendix. It would have been useful to have summaries in the main text...

_References:_

[A] Chen, Bingqing, et al. "Enforcing policy feasibility constraints through differentiable projection for energy optimization." Proceedings of the Twelfth ACM International Conference on Future Energy Systems. 2021.

[B] Sanket, Shah, et al. "Solving online threat screening games using constrained action space reinforcement learning." Proceedings of the AAAI Conference on Artificial Intelligence. Vol. 34. No. 02. 2020.

**Questions:**

1. In Table 3, the difference between 2S and the other methods *decreases* as the penalty factor increases. Why is that? I would have expected the opposite (as in Tables 2 and 4).
1. On line 355 the paper says, "On the other hand, the advantage of our 2S method over other approaches actually becomes more significant as the capacity increases, demonstrating the superior accuracy of our approach." However, the difference between 2S and other approaches for the lowest penalty factor actually decreases in absolute value?

**Limitations:**

The limitations are not described, despite having responded 'yes'. The fact that the paper does not acknowledge any of these limitations highlighted in the `weaknesses' section is worrying.

---

> ### Author Rebuttal · Authors · 2023-08-10
>
> Thank you for your positive review of our work.
> We also appreciate the points you raised in the weaknesses section.
> Please see our response below.
> The paper is also further strengthened now by our additional experiments in the overall response.
> We hope to further convince you of the merits of our work; please let us know if you have any additional questions.
>
> ---
> Response to "weaknesses":
>
> 1. Linear penalties: thanks for pointing out that Hu et al. can handle non-linear penalties; we will add a comment to clarify in the paper. We also want to point out that our framework, as stated, can already handle special cases of non-linearity: for example, one can (in many cases) express the absolute-value function in linear programs. In addition, the Section 4/Appendix B gradient calculations can in fact be adapted to handle general differentiable non-linear objectives just like Hu et al., though of course the caveat that the second-stage optimization problem needs to be solvable efficiently for the framework to be useful. We chose to present only MILPs as a main overarching application for this paper mainly because of their widespread use in discrete optimization, with readily available solvers.
>
> 2. Yes, training for 2S is slower than IntOpt-C, but as we pointed out in the overall response, the runtimes are quite comparable.
>
> 3. The projection of [B] is identical to the correction function proposed by Hu et al. The projection of [A] on the other hand is $\ell_2$ projection, which we hadn't compared against. For this rebuttal, we ran an experiment analogous to Table 1 in the main paper, for the alloy production problem (a covering LP) and the $\ell_2$ projection. We find that the $\ell_2$ projection performs even worse than the [B]/Hu et al. correction, for the linear penalty function used in the alloy production problem. Please see the overall response for more details.
>
> 4. We did struggle with the page limit. If accepted, we will move more information to the main body of the paper, given the extra page available.
>
> ---
> **Q1**: In Table 3, we omitted to present penalty factors $\ge 1$ for space reasons. Additionally, in that problem setting (proxy buyer knapsack), large penalty factors are unrealistic anyway. We now include the corresponding rows here.
>
> | PReg    | Penalty factor |     2S     |    Ridge   |    k-NN    |    CART    |     RF     |     NN     |     TOV    |
> |---------|:--------------:|:----------:|:----------:|:----------:|:----------:|:----------:|:----------:|:----------:|
> |         |        1       | 10.90±0.15 | 10.93±0.19 | 11.11±0.17 | 11.16±0.14 | 11.01±0.31 | 11.26±0.23 |            |
> | cap=100 |        2       | 12.31±0.16 | 12.48±0.20 | 12.49±0.21 | 13.77±0.26 | 12.60±0.39 | 12.78±0.30 | 29.68±0.14 |
> |         |        4       | 14.54±0.15 | 15.57±0.25 | 15.68±0.39 | 19.01±0.56 | 15.77±0.62 | 15.84±0.50 |            |
> |         |        1       | 10.23±0.12 | 10.46±0.23 | 10.40±0.18 | 10.46±0.19 | 10.49±0.21 | 10.86±0.30 |            |
> | cap=150 |        2       | 11.18±0.15 | 11.88±0.30 | 11.63±0.20 | 12.56±0.31 | 11.83±0.19 | 12.12±0.17 | 40.23±0.19 |
> |         |        4       | 13.20±0.16 | 14.71±0.49 | 14.43±0.33 | 16.75±0.63 | 14.53±0.29 | 14.65±0.41 |            |
> |         |        1       |  6.77±0.36 |  7.67±0.18 |  7.51±0.27 |  7.71±0.20 |  7.67±0.16 |  8.00±0.65 |            |
> | cap=200 |        2       |  8.19±0.12 |  8.84±0.22 |  8.69±0.26 |  9.24±0.30 |  8.80±0.20 |  8.97±0.37 | 48.13±0.24 |
> |         |        4       |  9.71±0.35 | 11.17±0.40 | 11.06±0.32 | 12.29±0.59 | 11.05±0.46 | 10.91±0.53 |            |
> |         |        1       |  1.37±0.08 |  3.08±0.19 |  2.94±0.16 |  3.17±0.17 |  3.05±0.25 |  3.28±0.96 |            |
> | cap=250 |        2       |  3.34±0.15 |  3.80±0.20 |  3.73±0.15 |  3.94±0.20 |  3.79±0.26 |  3.89±0.58 | 53.43±0.26 |
> |         |        4       |  4.46±0.09 |  5.25±0.35 |  5.32±0.27 |  5.47±0.35 |  5.29±0.48 |  5.11±0.39 |            |
>
> Along with these extra rows, we can see that the trend of Table 3, in terms of the difference between 2S and other methods, is in fact identical to the trend in Table 2. The difference first decreases, then increases, as the penalty factor increases. We can explain this phenomenon as follows.
>
> First, when the penalty factor is small, the rational behavior for the buyer is to just take every order, and only decide which orders to drop when the true parameters are revealed (at close to no cost). 2S identifies and exploits this behavior for small penalty, while classic regression methods are agnostic to this possible tactic. Thus, the advantage of 2S compared to classic regression methods is large in the small penalty case.
>
> Second, when the penalty factor is large, 2S will analogously learn to be conservative, such that the first stage solution likely remains feasible under the true parameters, in order to avoid the necessary (and high) penalty due to having to change to a feasible solution. Again, classic regression methods will be agnostic to this possible tactic, leading to a large advantage of 2S over the classic methods.
>
> Table 4 only has the increasing trend from the large penalty, since it is neither a covering nor a packing program, and so there is no analogous tactic/exploitation for small penalty.
>
> **Q2**: Here, "advantage" refers to the improvement in *percentage* of our method over other approaches.
> Take penalty factor = 0.05 as an example, the improvement percentage is (8.67 - 1.26)\%/8.67\% = 85.47\% when capacity is 100, 91.37\% when capacity is 200, 94.73\% when capacity is 300, and 96.79\% when capacity is 400.
> We will clarify in the paper.
>
> ---
> Limitations: as discussed above, we will clarify the points you raised in the paper. Thank you for pointing them out.

---

> > ### Comment · Reviewer_Pw5Y · 2023-08-12
> > **Response to Rebuttal**
> >
> > Thank you for your response, esp., for running extra experiments with the L2 penalty (very interesting to know that it does worse!) and explaining the results in Table 3. I am still slightly on the fence about whether the proposed method is *always* better than Hu et. al. (e.g., if solving the problem with nonlinear penalties is too expensive), but I hope that the authors add this to the limitations section. Overall, I'm satisfied with the authors' response to my concerns and have increased my score to a 7.

---

> > > ### Author Response · Authors · 2023-08-13
> > > **Official Comment by Authors**
> > >
> > > We thank the reviewer for their further comments and improved evaluation of our work. If accepted, we will add a "Limitations" section in the paper to discuss the points raised in the reviews and our rebuttals.

---

### Official Review · Reviewer_Xm8w · 2023-07-06

**Soundness:** 3 good
**Presentation:** 3 good
**Contribution:** 3 good
**Rating:** 7
**Confidence:** 4

**Summary:**

The authors propose a framework for learning latent variables in optimization problems that appear both in the constraints and objective. In this setting, the user is given features and asked to provide a solution to an optimization problem where the objective and constraints of the optimization problem are partially observed and related to the features. Additionally, the user can optimally modify the solution once the true parameters are known at a cost based on the change. The overall goal is to ensure that the total regret is low where the regret is the total value of the final solution after fixing minus the fixing cost and minus the objective value of the optimal solution in hindsight. The authors propose that the second stage solution should be considered the output of an optimization problem which is given the first stage solution as input and then backpropagates through both optimization problems to update the weights of the predictive model predicting parameters for the first stage optimization problem. The authors propose differentiating through continuous relaxations of these optimization problems using previous work that differentiates through iterates of an interior point method.

The authors evaluate their approach on several settings to demonstrate improved predictive performance over the investigates baselines.


**Strengths:**

The main strength of the work is that it considers penalizing the recourse using a flexible optimization problem rather than having a domain-dependent method of doing so as was done in previous work for packing and covering. Additionally, the paper itself is easy to read and

**Weaknesses:**

Given that the main contribution of this work is that the framework has good empirical performance, it would be good to strengthen the experiments by evaluating against relevant baselines.

The work does not compare against relevant baselines and claiming generality to MILP to discount several differentiable continuous optimization baselines when in practice, the proposed approach simply relaxes the integrality constraints to consider differentiating through a continuous LP. Given that this approach considers differentiation of MILP with respect to the constraints as simply differentiating the constraints of the LP relaxation, the authors should evaluate against methods which can differentiate through constraints of an LP which includes cvxpylayers [1], and OptNet with quadratic regularization to differentiate through LP [2,3].

Additionally, the previous CombOptNet work [19 in the paper] which is formulated explicitly for learning constraints in combinatorial settings, and whose datasets are used for two of the three settings, is not compared against.


[1] Agrawal, Akshay, et al. "Differentiable convex optimization layers." Advances in neural information processing systems 32 (2019).
[2] Amos, Brandon, and J. Zico Kolter. "Optnet: Differentiable optimization as a layer in neural networks." International Conference on Machine Learning. PMLR, 2017.
[3] Wilder, Bryan, Bistra Dilkina, and Milind Tambe. "Melding the data-decisions pipeline: Decision-focused learning for combinatorial optimization." Proceedings of the AAAI Conference on Artificial Intelligence. Vol. 33. No. 01. 2019.


**Questions:**

Are there any specific reasons that previous work that learns constraints for LP or comboptnet are not applicable to the investigated settings?

Are there any specific components of this method which are specialized to handle integrality that cannot be handled by simply applying previous approaches for differentiating through continuous problems?

It might be helpful to compare the gradients of this approach on a subjective level as well. In equation (3) are the gradients from one component relatively large compared to gradients from the other? How do they differ overall from previous work?


**Limitations:**

Relevant limitations are addressed.

---

> ### Author Rebuttal · Authors · 2023-08-10
>
> Thank you for pushing us on running more experiments and comparing with more baseline methods, which strengthens the paper. We believe we have adequately addressed your concerns through the additional experiments presented in the overall response, along with the following discussion.
> We are happy to answer any additional questions you may have.
>
> -----
>
> Response to "weaknesses": originally, we didn't compare with cvxpylayers because cvxpylayers is a conic generalization of OptNet, and Mandi and Guns had already given empirical evidence that their approach is better than QPTL (which uses OptNet for differentiating through programs). However, after you pointed out the lack of comparison, we ran additional experiments comparing 2S with using cvxpylayers (without regularization, with quadratic regularization and with log-barrier regularization) in place of our Section 4 gradient computation approach. We find that (see Tables 1 and 2 in new pdf) 2S offers at least as good solution quality while being around 30-50\% faster. Please see the overall response for more details and the precise description of the experiments.
>
> We also added comparisons to CombOptNet, which is not designed to learn solutions to have good post-hoc regret, but rather, to learn $\hat{x}$ that is close to the optimal $x^*$. Our additional experiments find that CombOptNet both has far inferior solution quality in post-hoc regret, takes far longer to train, and needs more data to have reasonable generalization. Please again see the overall response for more details.
>
> -----
>
> **Q1**:
> They are applicable and we now have further experiments, see above and overall response.
>
> **Q2**: No, and we added experiments comparing 2S (using the Section 4 gradient computations) with using cvxpylayers (with different regularizations). Please see the above.
>
> **Q3**: This is a very interesting question. We re-ran the experiments for 2S to investigate. First note that in Equation 3, the terms correspond to gradients coming from the second and first stages of optimization respectively, and they have common factors that we will ignore. We find the following pattern: the gradients due to the second stage optimization is small compared to the first stage analogue. Furthermore, as the training continues, the gap widens. The effect is especially pronounced when the penalty factor is large. This can be explained by the fact that, if the penalty factor is large, then the model has incentive to give predictions that yield an $x^\ast_1$ that will change minimally to $x^\ast_2$ in the second stage optimization. Thus, as we train more and more, we can expect the $\frac{\partial x^\ast_2}{\partial x^\ast_1}$ term to decrease in magnitude. The same effect persists, but in a less pronounced way, even when the penalty factor is smaller.

---

> > ### Comment · Reviewer_Xm8w · 2023-08-16
> > **followup**
> >
> > **literature**
> >
> > With regards to the related literature, it seems there is more work on making predictions for unknown parameters in the constraints [1]. Additionally there is followup work from Hu et al for prediction in constraints [2], although I believe it was released after the neurips submission deadline.
> >
> > I agree that this space has been much less investigated than predicting hidden objective coefficients. However, it seems that there has been work regarding constraint prediction + optimization, although it is unclear how directly applicable these approaches are to this work. It would be helpful to tie in more related work to explain other methods that are in a similar vein to this approach and why the are or are not applicable. I believe this might also help with the literature weakness mentioned by Reviewer FnYQ. Overall, a more comprehensive explanation of related work not only serves to explain what methods inspired the proposed approach, but also to help readers understand how this approach compares to existing work when deciding whether it is suitable for their problem at hand.
> >
> >
> > For experiments against cvxpylayers, it is promising that the proposed approach gives similar or slightly improved performance. However, in my experience, Cvxpylayers is considerably slower than the QPTH code released with optnet due to implementation. As a result, the runtime improvements for this approach are unclear but in any case may be due to implementation details. Overall, it would be important to explain why the previous approaches perform well in terms of solution quality, and to either explain how a previously possible approach of applying optnet fail, or how the view taken by the proposed approach allows for new capabilities.
> >
> > [1] Nandwani, Yatin, Rishabh Ranjan, and Parag Singla. "A Solver-Free Framework for Scalable Learning in Neural ILP Architectures." Advances in Neural Information Processing Systems 35 (2022): 7972-7986.
> >
> > [2] Hu, Xinyi, Jasper CH Lee, and Jimmy HM Lee. "Branch & Learn with Post-hoc Correction for Predict+ Optimize with Unknown Parameters in Constraints." International Conference on Integration of Constraint Programming, Artificial Intelligence, and Operations Research. Cham: Springer Nature Switzerland, 2023.

---

> > > ### Author Response · Authors · 2023-08-17
> > >
> > > Thank you for your additional concrete and constructive comments. In view of your comments and other reviews, we have added a new overall response (see "official comment" to our overall rebuttal) to discuss at a high-level the related literature that isn't directly within the Predict+Optimize line of work. The one-line summary is that these works are either solving a different problem by design (though technically applicable, but will perform badly in our setting), or they are technical tools for differentiating through mathematical programs that are orthogonal to our new framework (the "Two-Stage"-ness with the post-hoc regret), which is our primary contribution. If accepted, we will incorporate this discussion into the paper. We understand your concern about further contextualizing our work, and we believe our new overall response addresses it.
> > >
> > > For the two additional references you pointed to, you are correct that the new Hu et al. [2] is very recent and concurrent work. This followup work still uses their prior "correction function" framework in AAAI'23, contrasting our new two-stage framework. As for the Nandwani et al. paper, as explained in our new overall response, it is designed to learn for a completely different goal, and is unlikely to work well for post-hoc regret. We have attempted for over 12 hours to run the knapsack benchmark on their implementation, but we had troubles setting up their environment.  Even their installation and demo code failed to run properly, as we tried doing this on multiple machines to make sure it is not a machine-specific problem. We have contacted the authors for help. If possible before the discussion deadline, we will endeavor to provide experimental results on the Nandwani et al. method, in addition to the CombOptNet results we already have, to further confirm our analysis.
> > >
> > > Please let us know whether we have addressed your concerns or if you have any remaining questions.
> > >
> > > -----
> > >
> > > Here we also respond directly to your comments about the implementation of cvxpylayers (and related tools such as QPTH).
> > > We agree that the runtime improvements may be due to implementation details, that QPTH (OptNet) for example might be a faster implementation for quadratic regularization.
> > > However, as our rebuttal experiments show, quadratic regularization gives *worse learning performance* than our implementation in terms of post-hoc regret (for small penalty, the difference is small, but the difference grows with the penalty).
> > > The learning performance, unlike runtime, is independent of the implementation.
> > > Our rebuttal experiments thus confirm that our choice of log-barrier regularization for its best learning performance, and we point out that OptNet supports *only* quadratic programs.
> > >
> > >
> > > As we point out in our overall responses, the primary contribution of our paper is in the framework. Tools for differentiating through LPs are just for *instantiating* our framework. Even **if** there are other tools that work better than our Appendix B calculations, these hypothetically better tools would not diminish our main contributions and should in fact further demonstrate the applicability of the framework.

---

> > > > ### Comment · Reviewer_Xm8w · 2023-08-17
> > > >
> > > > I thank the authors for better situating their contributions with respect to previous work in making predictions in the constraints both empirically and conceptually. Additionally, by providing context with respect to previous work, the authors better explain how their approach differs or why previous approaches are not applicable. Given the experiments demonstrating improved performance over previous approaches, and the fact that the authors consider their method flexible enough to handle alternate methods for differentiating through the "two-stage" problem, I believe that their contribution is now more substantially validated.

---

> > > > > ### Author Response · Authors · 2023-08-18
> > > > >
> > > > > We thank the reviewer again for the constructive discussion during this rebuttal period. We are particularly grateful for the push to compare with more methods and, more importantly, to further contextualize our work within decision-focused learning in general, including outside of Predict+Optimize. Both these changes substantially improve the coherence and significance of our message. We will incorporate the discussion raised in the reviews and our rebuttals into the final paper.

---

### Official Review · Reviewer_FnYQ · 2023-07-06

**Soundness:** 3 good
**Presentation:** 2 fair
**Contribution:** 2 fair
**Rating:** 4
**Confidence:** 4

**Summary:**

The authors develop a two-stage predict-and-optimize approach. There is a recent paper of Hu et al. [9] that extends the predict-and-optimize framework to having unknown parameters in the constraints. The authors argue that the Hu et al. approach, which requires defining both a "correction function" and a "penalty function", could be simplified by solving a single optimization problem. The authors show how to run the Mandi and Guns [15] solver to get the answer to this optimization problem. They then demonstrate a wide range of examples for their approach, including some real-world examples.

**Strengths:**

The authors have written a clear and easy-to-understand paper. I have not checked the proofs with great detail, but the mathematics looks very reasonable. The authors are getting impressive results on the examples that they explore.

The novelty is that the authors are combining the work of Hu et al. with the work of Mandi and Guns.

This is a fairly important problem within the Predict+Optimize framework and getting good solutions to these problems is important, so the significance is that this approach could hopefully be used. For the examples that the authors are demonstrating, their approach is clearly better to Hu et al.

**Weaknesses:**

As a point of order, it's frustrating that papers sometimes cite references so narrowly. I understand this a bit better when a paper is very mathematical, for instance building a proof based on another proof. But in this paper, the proofs are fairly algorithmic/arithmetic and involve well-known components. I haven't tried to discover who the authors are, but I notice that that 7 of the 23 references [1, 2, 3, 7, 14, 15, 16] are to the same group of close collaborators. Then, another 11 of the 23 references [6, 8, 11, 12, 13, 17, 18, 19, 20, 21, 22] are to software packages, test problems and text books. This leaves only 5 of 23 references which are to papers that are somehow intellectually related but not from a single group of collaborators (of these 5 remaining papers, two are to Elmachtoub and co-workers, two are to Hu and co-workers, and one is to Wilder et al.).

I am anxious about how many of the papers (7) refer to one closely-related group when only 5 of the cited papers are to work outside the group (I discard the 11 software packages / textbooks / test instances because these don't require the same work in linking ideas). Work that singularly focuses on just a single small group of collaborators sometimes neglects to take the broad perspective that often characterizes good science.

The reason that I'm marking "Fair" in the "Presentation" tick mark is this very limited interaction with the literature. Other than this problem, the paper is quite clear.

****

At least as the paper is worded, there is a Hu et al. [9] framework that has been introduced that matches the authors setting, but everything would be better if only we would "adapt the approach of Mandi and Guns [15]" [Section 1, line 79]. For example, most of Section 4 (lines 211 - 268) discuss an adaptation of Mandi and Guns [15]. The point of the paper often seems to be that if only Hu et al. had read Mandi and Guns a bit more carefully, Hu et al. would have done things differently.

Overall, the current paper reads like a "letter to the Editor" objecting to elements of the work of Hu et al. [9] and trying to correct it. The Hu et al. framework is described in detail (lines 40-82 in the introduction and also Section 3 seem dedicated to Hu et al. and why what they're doing is wrong). Then, Section 4 proposes fixes for the Hu et al. framework using the Mandi and Guns framework.

The impression this leaves is that the authors' work is a bit incremental. As the authors acknowledge, their framework and the framework of Hu et al. are "mathematically equivalent in expressiveness". So the only difference is that the Hu et al. [9] paper develops a "correction function" and a "penalty function" whereas these authors solve an additional optimization problem (this optimization problem seems very similar to the one already proposed in Mandi and Guns).

The reason that I'm marking the "Contribution" as "Fair" is that I can't see anything except modifying Hu et al. to work more like Mandi and Guns.

****

I disagree with the authors that their approach "should be the canonical framework for the Predict+Optimize setting". The authors have a lot of impressive results, especially in comparison to Hu et al., but there are times where solving an optimization problem may take too long and it may be better to have a function that is quick-to-evaluate.

This small note (that the authors are claiming that it's always better to solve an optimization problem) is the only reason that I'm marking the "Soundness" as "Good" rather than "Excellent".

**Questions:**

I've tried to be pretty specific as to what are the weaknesses of the paper so that the authors can correct as relevant. Please take the "Weaknesses" section as a set of questions where I'm happy to be corrected.

**Limitations:**

This part of the paper is fine.

---

> ### Author Rebuttal · Authors · 2023-08-10
>
> Thank you for your comments and detailed review of our work. Below, we address the points you raise in "weaknesses".
>
> **Significance**: We believe there might be a misunderstanding of the main message of our work. Our point is not "everything is better if we do it the way of Mandi and Guns". In fact, Hu et al. also use the work of Mandi and Guns as a technical component for differentiating through LPs. As we argue, however, their high-level view of Predict+Optimize with unknowns in constraints, by imposing a correction function, is sub-optimal. As a result, they're using Mandi and Guns's work not in the best way possible. We give a simple change in perspective, which enables better algorithmic use of differentiating through LPs.
> Please also refer to "main message" in our overall response for more details.
> We also point out that, at least in our opinion (and a view that seems shared also by reviewers Pw5Y and Jj2a), a simple idea that leads to much better performance is valuable, instead of a downside on the quality of the work.
> The fact that we presented a detailed comparison with the Hu et al. framework should also be a plus and not a minus.
>
> **Canonicity** of the framework: we agree that there is a tradeoff in runtime and learning capabilities/solution quality. We will tone down the claim in the paper by remarking *up front* the tradeoffs. Thank you for pointing out this nuance that we forgot to address. Please also see our overall response concerning the training times of these different approaches.
>
> **Literature**: While we have cited all the works that directly inspire our paper, we understand that we have indeed missed some related references. We are happy to incorporate and cite other missing works, and welcome additional suggestions. We additionally wish to point out that, as far as we understand, the Stuckey group and the Guns group are two distinct research groups on Predict+Optimize, despite Guns occasionally collaborating with the Stuckey group. The focuses on the groups also seem somewhat different: the Stuckey group seems to work more directly on combinatorial optimization whereas the Guns group works more on "continuous" optimization techniques as far as we can tell.

---

> > ### Comment · Reviewer_FnYQ · 2023-08-11
> > **Thanks for the response**
> >
> > **Significance.** Agreed that the detailed comparison to the Hu et al. framework is not a weakness, the issue is that just adjusting one paper (Hu et al.) to work more like another paper (Mandi & Guns) seems insufficient for a NeurIPS publication, especially since the Mandi & Guns paper seems to be among the 7 papers from a close group of collaborators where the authors focus their attention. If the authors had considered the literature more broadly before deciding on the Mandi & Guns paper to modify the Hu et al. framework, this would be more manageable.
> >
> > **Canonicity.** Thanks to the authors for agreeing to tone down the claim.
> >
> > **Literature.** Thanks to the authors for acknowledging that they have missed some papers and for promising to incorporate them. I don't have knowledge on the relationship between research groups in this area, I've only observed that the 7 papers identified share a lot of the same authors.

---

> > > ### Author Response · Authors · 2023-08-12
> > >
> > > We thank the reviewer for further engaging.
> > >
> > > **Significance**:
> > > We believe that there is still a misunderstanding of the contribution of our paper.
> > > The point of our work is **not** to "work more like Mandi and Guns".
> > > As explained in our overall response, our main contribution is the new Predict+Optimize framework for handling unknown constraints.
> > > The learning method we propose, in order to substantiate our new framework, is one for training a neural network.
> > > As such, it requires a way to compute training gradients, and in fact so does the training method proposed by Hu et al. for their prior framework.
> > > Both papers chose the machinery provided by Mandi and Guns only as *a technical tool* for differentiating through LPs.
> > > Thus, the improvement of our paper over Hu et al. is **not** in making things "more like Mandi and Guns", but in **how** we use those tools (that we use it *also* for a second-stage optimization problem).
> > > As the reviewer already noted, we get impressive empirical results from this change of perspective.
> > >
> > > In our rebuttal to reviewer Xm8w, we explained that we chose Mandi and Guns because their work had already compared with QPTL (LP+quadratic regularization using OptNet, and cvxpylayers is just a conic extension to OptNet).
> > > Our choice was therefore **principled**, and not because we only "focus [our] attention on a close group of collaborators".
> > > In the rebuttal experiments, we have further demonstrated that this choice is indeed the best, with our 2S method outperforming all the reviewers-suggested alternatives, including cvxpylayers and CombOptNet.
> > >
> > > We also want to re-emphasize that Hu et al. was the **first** and **only** paper to date to propose a Predict+Optimize framework that handles unknown parameters in constraints. Naturally, our paper explains how we are different from and how we improve upon that work.
> > > We find a *simple* idea (which, again, is *not* "be more like Mandi and Guns") to significantly improve the performance, and this should be seen as a benefit and not something to be considered incremental.
> > >
> > > Finally, as the reviewer recognized, both Predict+Optimize and specifically the problem of handling unknowns in constraints, are important research directions for the NeurIPS/ICML community.
> > > Given that our paper gives a simple idea which yields substantial empirical improvements, as verified both by the in-paper and rebuttal experiments, we strongly believe that our work will be a significant and valuable contribution to the community.
> > >
> > >
> > > **Literature**: As we previously explained in the rebuttal, the Stuckey Group and the Guns Group are distinct. Both groups have made significant contributions to the Predict+Optimize area, and have published many related works. Given that the main theme of the paper is Predict+Optimize, we don't understand the issue of citing these groups. If the reviewer is aware of additional missing references, we are more than happy to cite them.

---

> > > > ### Comment · Reviewer_FnYQ · 2023-08-18
> > > > **Thanks**
> > > >
> > > > I really appreciate the updated literature review response in the overall comment. This helps things. As the authors have observed, I like large parts of this paper and the additional literature review helps.

---

> > > > > ### Author Response · Authors · 2023-08-18
> > > > >
> > > > > We are glad that you appreciate our additional literature review. If there are any concrete concerns/suggestions that we can address for you to revise your evaluation, please let us know.

---

### Author Rebuttal · Authors · 2023-08-10

Thank you for your constructive and in-depth feedback for improving the paper.
We are encouraged by the reviewers recognizing that our paper 1) tackles an important problem (reviewers FnYQ, Jj2a) in Predict+Optimize, 2) presents a simple/sensible, elegant and flexible solution (reviewers Pw5Y, Jj2a, Xm8w), and 3) gives experiments that are close to real-life (reviewers FnYQ, Jj2a).
We are also grateful for the constructive criticisms which can improve our paper.
In this overall response, we wish to emphasize again the main conceptual contribution of this paper, as well as address the criticisms on our experiments and citations.
We will additionally respond to remarks and questions separately for each individual review.

We believe that our rebuttals below have adequately addressed your criticisms, and hope that you will improve your evaluation of our work based on these responses. Please let us know if you have remaining concerns. We will address them.

**Main message**: While Hu et al. gave the first Predict+Optimize framework to handle unknowns in constraints, their framework required specifying an ad-hoc correction/recourse/differentiable projection. In this work, we provide a *simple* change of perspective (Section 3), viewing the recourse action itself *naturally* as the solution of an optimization problem. This simple change (a) yields much better test-time performance (see e.g. experiments in Table 1 in the paper), (b) allows for post-hoc correction even when the stage 1 solution (a soft commitment) doesn't violate constraints under the true parameters (as was recognized by reviewer Pw5Y), and (c) enables the algorithmic training methodology for generalizing to handle MILPs in Section 4. We emphasize that, especially in the context of frameworks, simplicity and the associated flexibility is a virtue and not a downside (as was recognized by reviewers Pw5Y, Jj2a, Xm8w).

**Experiments**: Thank you for pushing us on additional experiments, which strengthen the empirical part of our paper. Based on reviewer feedback, we ran two new experiments comparing with the suggested methods.

1. We applied cvxpylayers and CombOptNet to the 0-1 knapsack benchmark to compare with our 2S method. See Table 1 in the new pdf for post-hoc regrets and Table 2 for the training times. More precisely, for cvxpylayers, we use it with various regularizations (a. LP with no regularization, b. with quadratic regularization, c. with log-barrier as in our paper) to replace the Section 4/Appendix B gradient calculations. For CombOptNet, we just run it as is, since it is a method for learning unknowns in constraints. These methods are evaluated at test-time using the Two-Stage framework, as we did in the paper.\
We find that cvxpylayers never gives better solution quality while 2S is 30\%--50\% faster. For CompOptNet, the solution quality is very bad, since it was designed to learn a first-stage solution $\hat{x}$ close to $x^\ast$ and not learning for small post-hoc regret. CompOptNet is also drastically slower. We further observe that, using only 700 training samples (as in the experiments in the paper and rebuttal), CompOptNet does not have good generalization. Only with the full 4500 training samples in the CombOptNet paper do we get some reasonable generalization. See Figures 1-4 for training+test loss curves (using loss $\|\hat{x}-x^\ast\|^2$), for 700 training samples (Figures 1,2) and 4500 training samples (Figures 3,4). This is evidence that CompOptNet is more data-hungry than our 2S method.

2. We tested the differentiable projection idea in references [A,B] given by reviewer Pw5Y, on the Alloy Production benchmark. The projection in [B] is identical to the Hu et al. correction function, and so we only additionally tested the $\ell_2$ projection method in [A], implemented using cvxpylayers. The experiment set-up follows that of Table 1 in the submission: both training and testing use $\ell_2$ projection in the second stage, as opposed to solving the second stage optimization problem defined in Section 3. Table 3 in the new pdf shows both the post-hoc regret and training time for $\ell_2$ projection. We find that, not only is $\ell_2$ projection slow, but it has even worse post-hoc regret than the Hu et al. correction. We suspect that this is due to the Hu et al. correction function preserving the direction of the solution vector whereas $\ell_2$ projection can change the direction, and that this makes a difference for Alloy Production. In any case, this experiment confirms again that our Two-Stage framework has better post-hoc regret than a framework based on differentiable projections, reinforcing the main message of our paper.

**Runtime**: Appendix E gives the training time for each method tested in the submission. In the Alloy Production problem, which is the *only* setting where the Hu et al. IntOpt-C method can be applied, their running time is quite comparable with our proposed 2S method. In the additional experiments in the rebuttal (see Tables 2,3 in the new pdf), the 2S method is also faster to run than cvxpylayers and CombOptNet, while offering at least as good and sometimes substantially better learning performance. We believe that our method presents a reasonable tradeoff between runtime and learning in practice. If accepted, we will use the extra page to include a runtime discussion in the main paper.

**Literature**: We thank the reviewers for the additional references. The Predict+Optimize literature, and more generally the decision-aware learning literature, is rapidly growing in recent years, with different research groups even sometimes calling the same concepts different names. As such, it has become increasingly difficult to keep track of all the relevant literature. While we have cited all the works that directly inspire ours, we have nonetheless missed some other. We are grateful for the reviewers pointing us to works we have missed, and will address them in the paper.

---

> ### Author Response · Authors · 2023-08-17
>
> **Further literature response**
>
> Our paper should be placed in the context of Predict+Optimize intersecting with learning unknowns in constraints.  In this space, the *only* prior framework is by Hu et al. (cited work, and a very recent followup work after the NeurIPS deadline).  In view of reviewers' comments, here we summarize works in "non-Predict+Optimize" decision-aware learning, in particular those related to learning unknowns in constraints.  If accepted, we will incorporate this discussion into the paper, to further contextualize our work.
>
> These works can be placed into two categories:
>
> 1. *Papers proposing methods for learning unknowns in constraints, but with very different goals and measures of loss.*
> For example, CombOptNet and Nandwani et al. focus on learning parameters so as to make the predicted optimal solution (first-stage solution in our framework) as close to the true optimal solution $x^*$ as possible in the solution space/metric.
> By contrast, our paper explicitly formulates the two-stage framework and post-hoc regret in order to directly capture rewards and costs in application scenarios.
> As one would expect, and as verified by our rebuttal experiments, these other methods yield worse predictive performance when evaluated on the post-hoc regret, under our two-stage framework.
>
> 2. *Papers that give ways of differentiating through LPs or LPs with regularizations, as a technical component in a training algorithm.* These works are only tools, and related to our technical calculations in Section 4 and Appendix B (as a specific implementation of our framework). As such, these tools can be used in place of our Section 4/Appendix B. However, we point out that: (i) these tools are essentially *orthogonal* to our primary contribution, which is the two-stage *framework* and *how* we use technical tools to differentiate through LPs, and not *which* tools we use, and (ii) nonetheless, our rebuttal experiments demonstrate that our Appendix B tool performs at least as well in post-hoc regret performance as the others, while being faster.
> More specifically, we compared with cvxpylayers with a) no regularization, b) quadratic regularization and c) log-barrier (like our Section 4/Appendix B).
> The key indicator of its predictive performance is the type of regularization used, with the log-barrier version performing the best, but still slightly worse than our method. Reviewers also mentioned OptNet, which we did not explicitly compare with in terms of runtime, but we point out that OptNet only supports quadratic programs and so can only support quadratic regularization. OptNet and cvxpylayers use the same approach for quadratic programs, and so their difference would only be in the runtime.

---

### Decision · Program_Chairs · 2023-09-21

**Decision:**

Accept (poster)

**Comment:**

The reviewers agree that the paper addresses an important problem (unknown parameters in the constraints) in Predict+Optimize with thorough experiments including detailed ablations and comparison with multiple baselines. The idea is also quite nice, paper well-written and easy to follow. The negative reviewer also agreed that the concerns regarding to literatures are also addressed.